# CDK1 phosphorylates WRN at collapsed replication forks

Valentina Palermo[1], Sara Rinalducci[2], Massimo Sanchez[3], Francesca Grillini[1], Joshua A. Sommers[4], Robert M. Brosh Jr[4], Lello Zolla[2], Annapaola Franchitto[5] & Pietro Pichierri[1]

Regulation of end-processing is critical for accurate repair and to switch between homologous recombination (HR) and non-homologous end joining (NHEJ). End resection is a two-stage process but very little is known about regulation of the long-range resection, especially in humans. WRN participates in one of the two alternative long-range resection pathways mediated by DNA2 or EXO1. Here we demonstrate that phosphorylation of WRN by CDK1 is essential to perform DNA2-dependent end resection at replication-related DSBs, promoting HR, replication recovery and chromosome stability. Mechanistically, S1133 phosphorylation of WRN is dispensable for relocalization in foci but is involved in the interaction with the MRE11 complex. Loss of WRN phosphorylation negatively affects MRE11 foci formation and acts in a dominant negative manner to prevent long-range resection altogether, thereby licensing NHEJ at collapsed forks. Collectively, we unveil a CDK1-dependent regulation of the WRN-DNA2-mediated resection and identify an undescribed function of WRN as a DSB repair pathway switch.

[1] Section of Experimental and Computational Carcinogenesis, Department of Environment and Health, Istituto Superiore di Sanità, Rome 00161, Italy. [2] Proteomics Lab, Department of Ecology and Biology, Università della Tuscia, 01100 Viterbo, Italy. [3] Section of Gene and Cell Therapy, Department of Neurosciences, Istituto Superiore di Sanità, 00161 Rome, Italy. [4] Laboratory of Molecular Gerontology, National Institute on Aging, NIH, NIH Biomedical Research Center, Baltimore, Maryland 21224, USA. [5] Section of Molecular Epidemiology, Department of Environment and Health, Istituto Superiore di Sanità, 00161 Rome, Italy. Correspondence and requests for materials should be addressed to P.P. (email: pietro.pichierri@iss.it).

DNA double-strand breaks (DSBs) occurring during DNA replication are the most harmful DNA lesions, as they are direct precursors of chromosome rearrangements commonly found in cancer[1–3]. Moreover, replication-dependent DSBs are often induced by anticancer drugs, such as the topoisomerase I poison camptothecin (CPT) or its derivatives[4]. To deal with DSBs, cells evolved two distinct DNA repair mechanisms: non-homologous end joining (NHEJ) and homologous recombination (HR)[5,6]. NHEJ operates independently of homology at DNA ends, whereas HR uses the undamaged sister chromatid as template to reconstitute the integrity of DNA[5,7]. Thus, HR is the pathway of choice to repair the replication-dependent DSBs, as NHEJ in the S-phase may incorrectly rejoin the one-ended DSB with another non-homologous DNA end from a different chromosome leading to complex chromosome exchanges[7].

The crucial step for HR repair is the resection taking place at DNA ends, which provides the kb-long single-stranded DNA (ssDNA) tails required for RAD51 binding and subsequent strand invasion[8–10]. Studies from yeast demonstrated that end resection begins with limited processing by the Mre11-Rad50-Xrs2 complex (MRE11 complex) in cooperation with Sae2, followed by long-range resection performed by two independent and alternative pathways, consisting of Exo1 or the RecQ helicase Sgs1 in combination with Dna2 (refs 11,12). Several studies demonstrated that the DNA repair pathway choice between HR and NHEJ is regulated by the extensive end resection, which contributes to the displacement of NHEJ factors from DSBs shifting the balance to HR[8,9]. Furthermore, to prevent unscheduled HR or NHEJ activity, cyclin-dependent kinases (CDKs) supervise activity and function of end-resection factors[12,13]. Although it is well established that CDKs regulate the initial steps of the end-resection process, less is known about the regulation of long-range end resection[8,13], especially in human cells. Indeed, the initial stages of end resection are fairly well conserved in human cells, where the Sae2-orthologue, CtIP and the MRE11 complex are crucial during the initial steps of end resection, which is regulated by CDKs[12]. In human cells, CDKs directly regulate EXO1 (ref. 14); however, whether they also regulate the DNA2-dependent long-range resection pathway is unknown. Moreover, it is still debated whether DNA2 acts preferentially with WRN or BLM during long-range resection[15–17]. Although a recent work in human cells demonstrated that DNA2 can indifferently perform end resection with WRN or BLM, the relevance of WRN for the correct execution of end resection is difficult to appreciate because of pleiotropic functions of the protein and compensatory activity of EXO1 (refs 9,17–19). WRN is an intriguing candidate for CDK-dependent regulation of the DNA2 branch of long-range end resection. Indeed, WRN undergoes multiple phosphorylation events in response to DNA damage or replication stress and associates with the MRE11 complex[19–22]. Moreover, loss of WRN markedly sensitizes cells to CPT, an agent that induces fork collapse resulting in DSBs in S-phase[23,24]. Therefore, WRN may play a prominent role to regulate end resection at replication-dependent DSBs.

Here we describe CDK1-mediated phosphorylation of WRN, which represents a novel mechanism by which CDK1 controls long-range end resection and DNA repair pathway choice at replication-dependent DSBs in human cells. Abrogation of WRN phosphorylation impinges on proper MRE11 recruitment, stimulating NHEJ repair of replication-dependent DSBs and enhancing genome instability.

## Results

**WRN is phosphorylated at S1133 by CDKs after DNA damage.** Resection of DNA ends at DSBs is regulated by CDKs[13]. The WRN protein is critical for the response to replication-dependent DSBs, interacts with the MRE11 complex and collaborates with DNA2 in the end resection, and is phosphorylated by ATM[17,20,22]. Thus, WRN is a strong candidate for a CDK-dependent regulation in the context of HR-mediated DSBs repair.

Analysis of WRN revealed the presence of two N-terminal minimal (S/TP) CDK consensus sequences (HLS$^{426}$PND and HLS$^{453}$PND), and of a single, optimal, putative CDK phosphorylation site in the C-terminal region (S$^{1133}$PEK—S/TPxR/K). To determine whether these residues could be phosphorylated *in vivo*, we transiently over-expressed a Flag-tagged wild-type form of WRN in HEK293T cells, treated cultures with hydroxyurea (HU) to block replication or etoposide to induce DSBs, and performed mass spectrometry (MS) analysis on the immunopurified WRN protein (Fig. 1a). MS analysis of WRN phosphopeptides identified several phosphorylation sites and, among them, Serine 1133 (Fig. 1b). Interestingly, S1133 is conserved across species and was found to be phosphorylated also in the etoposide-treated cells (Fig. 1b).

To confirm that S1133 can be phosphorylated by CDKs, we performed an *in vitro* kinase assay using purified recombinant CDK2–CyclinA complex and the glutathione-S-transferase-fused 940–1,432 WRN fragment as substrate[20]. As shown in Supplementary Fig. 1, the C-terminal WRN fragment was readily phosphorylated by CDK2.

To determine if WRN S1133 could be phosphorylated by CDKs *in vivo*, we carried out IP-WB experiments using HEK293T cells transiently transfected with the Flag-WRN plasmid and exposed or not to the CDK inhibitor roscovitine in the last 4 h before sampling. To detect phosphorylation at S1133, we used a rabbit antibody directed against the CKSIMVQ{pS$^{1133}$}PEKAYSS peptide of WRN. The anti-pS1133WRN efficiently discriminated between the unphosphorylated and the phosphorylated WRN peptide, and easily detected the 940–1,432 WRN fragment phosphorylated *in vitro* with the CDK2–CyclinA complex (Supplementary Fig. 2A,B). When anti-Flag IPs were probed using the anti-pS1133WRN antibody, a highly evident immunoreactive band, corresponding to phosphorylated WRN, was detected, and its intensity was almost completely abrogated by the pan-specific CDK inhibitor roscovitine or by treatment of the IP with λ-phosphatase (Fig. 1c). Notably, immunoreactivity to the anti-pS1133WRN antibody was also reduced by treatment with a specific CDK1 inhibitor (RO-3306, CDKi) (Supplementary Fig. 3A), which decreased intensity of the WB signal similarly to roscovitine, suggesting that S1133 can be primarily targeted by CDK1 *in vivo*. To test if the anti-pS1133WRN antibody could actually mark S1133 modification *in vivo*, we carried out the IP-WB experiments in HEK293T cells over-expressing the Flag-WRN wild type or its S1133A unphosphorylable form. As expected, an anti-pS1133WRN-reactive band was detected in IPs from wild-type WRN-expressing cells and its intensity was reduced by the CDK1i (Fig. 1d). Interestingly, anti-pS1133WRN immunoreactivity was almost completely suppressed in IPs from cells expressing S1133A-WRN, and was comparable to that observed in wild-type IPs dephosphorylated by λ-phosphatase treatment (Fig. 1d). Moreover, anti-pS1133WRN signal was almost undetectable in IP from Werner's syndrome (WS) cells (Supplementary Fig. 3C).

Altogether, our results indicate that WRN is phosphorylated by CDK1 at S1133 in untreated cells and after DSB induction.

**WRN phosphorylation promotes long-range resection.** To explore whether phosphorylation by CDK1 regulated WRN role

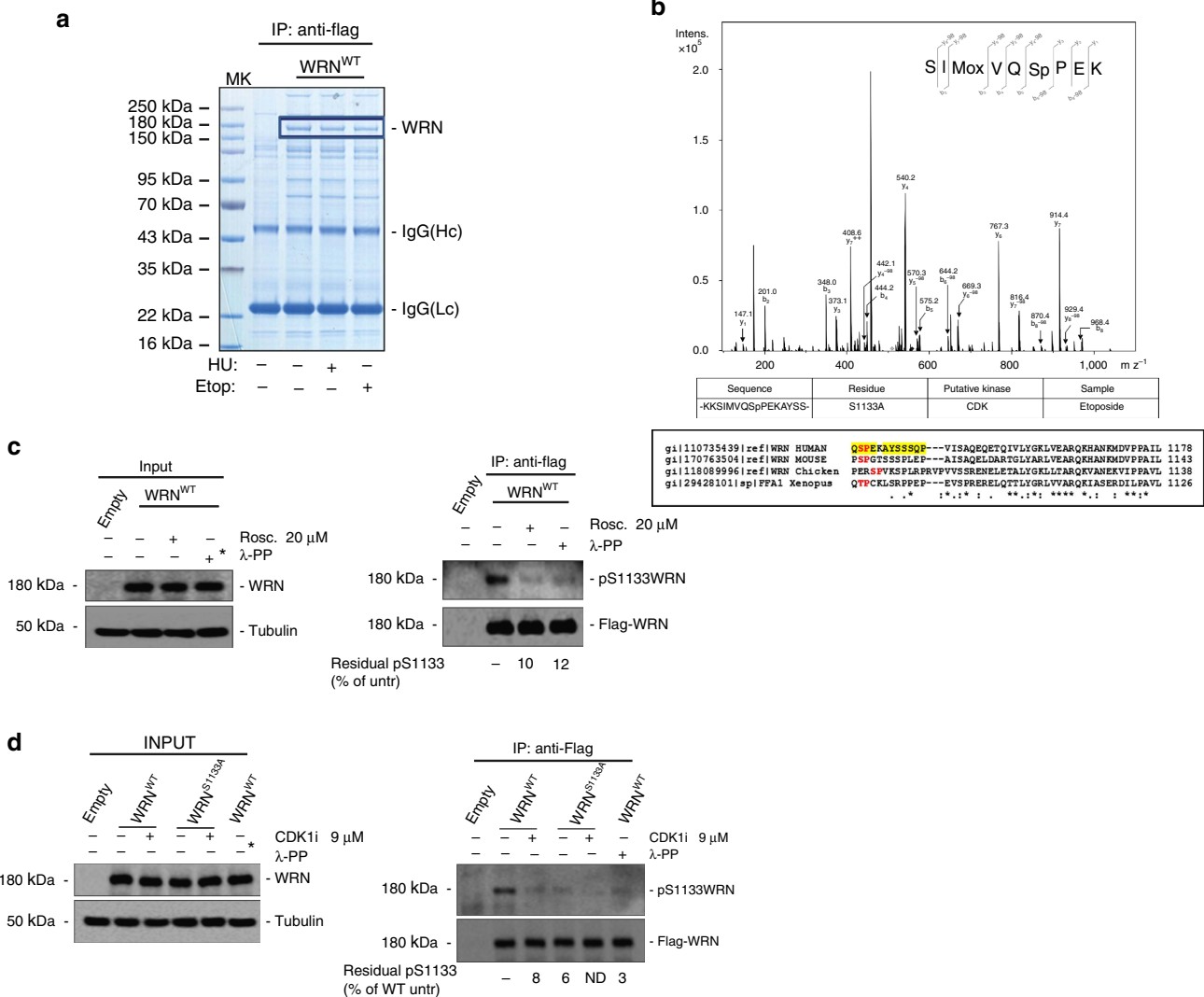

**Figure 1 | Identification of CDK phosphorylation sites in the WRN protein.** (**a**) HEK293T cells transiently transfected with pCMV-Flag-WT-WRN were treated with 500 nM etoposide (Etop) for 3 h, 2 mM HU for 8 h or nothing. Flag-WRN was immunoprecipitated (IP) with an anti-Flag antibody, separated by SDS–polyacrylamide gel electrophoresis (SDS–PAGE) and the bands corresponding to WRN (blue box) excised, digested and analysed by MS as shown in **b**. MS/MS analysis identified WRN S1133 phosphorylated from etoposide-treated sample. Multiple alignment of different WRN homologues shows that S1133 is highly conserved. (**c**) WRN is phosphorylated by CDKs *in vivo*. HEK293T cells transiently transfected with an empty vector or a vector expressing Flag-WT-WRN were treated or not with roscovitine for 4 h. WRN was IP with an anti-Flag antibody; 9/10 of IP were subjected to SDS–PAGE and detected by WB with the anti-pS1133WRN antibody, while 1/10 was detected by anti-WRN, as indicated. Where indicated, anti-Flag IP was treated with λ-phosphatase as an internal control. One-fiftieth of the lysate was blotted with an anti-WRN antibody to verify transfection. An anti-tubulin antibody was used as loading control. (**d**) Serine 1133 is targeted by CDK1. HEK293T cells transiently transfected as indicated were treated or not with RO-3306 (CDKi) for 4 h, IP with an anti-Flag antibody and subjected to WB as indicated in **c**. The 9/10 of IP were subjected to SDS–PAGE and detected by WB with the anti-pS1133WRN antibody, while 1/10 was detected by anti-WRN, as indicated. Asterisks showed in the input blots from **c** and **d** denote that the sample was not actually treated with λ-phosphatase but comes from the lysates used in the corresponding λ-phosphatase-treated IP reaction.

during end resection, we used WS-derived SV40-transformed fibroblasts, which does not show any detectable WRN protein, to generate cell lines expressing the wild type (WRN[WT]), the unphosphorylable (WRN[S1133A]) or the phosphomimetic mutant (WRN[S1133D]) form of WRN (Fig. 2a). As WS cells are hyper-sensitive to CPT treatment[23], end resection was analysed in the context of replication-dependent DSBs induced by micromolar doses of this drug. Of note, increased phosphorylation of WRN at S1133 was detected after CPT treatment (Supplementary Fig. 3B), and phosphorylation at S1133 was confirmed by MS in anti-Flag IPs from CPT-treated cells (Supplementary Fig. 4). To evaluate end resection, we directly quantified the generation of ssDNA by detecting 5-Iodo-deoxyuridine (IdU) foci by immunofluorescence

under non-denaturing conditions after pulse-labelling of nascent DNA. Using this IdU/ssDNA assay, we analysed the formation of ssDNA following 5′–3′ trimming of the parental DNA after induction of the one-ended DSB[25,26] (Fig. 2b). In untreated cells, no ssDNA was detectable irrespective of WRN phosphorylation (Fig. 2b and Supplementary Fig. 5A,B). At 30 min after CPT treatment, ssDNA begun to be detectable although the fluorescence intensity was low, which is consistent with the initial, short-range, end resection. However, no differences were observed among the wild-type and the WS-derived phosphomutant cells (Fig. 2b and Supplementary Fig. 5A,C). In contrast, robust accumulation of ssDNA was revealed after 90 min from treatment and it was significantly decreased in WS

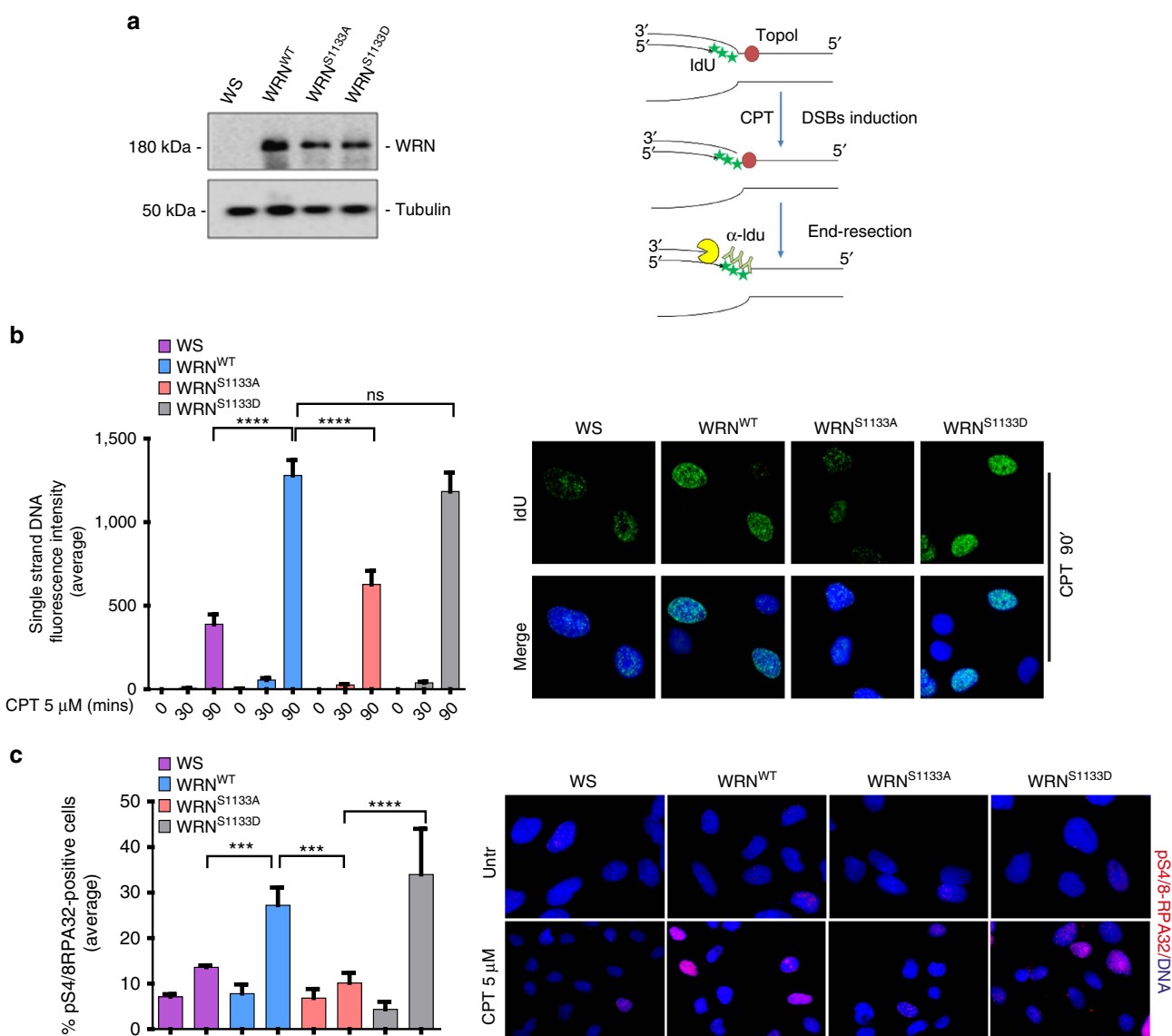

**Figure 2 | Phosphorylation of WRN by CDK1 promotes long-range resection at replication-dependent DSBs. (a)** Generation of WS-derived SV40-trasformed fibroblasts stably expressing the wild-type form of WRN, S1133A- or S1133D-WRN mutant. Western blotting shows WRN expression levels in each stable cell lines. **(b)** Evaluation of ssDNA by anti-IdU immunofluorescence under non-denaturing condition. The scheme depicts how ssDNA can be visualized at collapsed replication forks. CPT treatment results in one-ended DSBs at replication forks leading to 5′–3′ resection of template DNA by nucleases (pacman) thus exposing nascent ssDNA, which is detected by our IdU/ssDNA assay. For sake of clarity, only IdU-substituted nascent leading strand was represented. Nascent DNA was pre-labelled for 15 min with IdU before treatment and labelling remained during treatment with CPT as indicated. The graph shows the mean intensity of IdU/ssDNA staining measured from two independent experiments ($n = 100$, each biological replicate), data are presented as mean ± s.e.m. Representative images of IdU/ssDNA-stained cells treated with CPT for 90 min are presented in the panel. **(c)** Cells were treated with CPT as indicated and analysed for the formation of pS4/8-RPA32 foci by IF. The graph shows the percentage of pS4/8-RPA32 foci-positive cells as obtained from three independent experiments ($n = 150$, each biological replicate), data are presented as mean ± s.e.m. The panel shows representative images from untreated and CPT-treated cells. Statistical analysis was performed by the analysis of variance test (****$P < 0.001$; ***$P < 0.001$; NS, not significant).

or WRN[S1133A] cells (Fig. 2b and Supplementary Fig. 5A). Interestingly, expression of S1133A-WRN reduced ssDNA >70% compared with the wild-type or WRN[S1133D] cells (Fig. 2b). During end resection, ATM or DNA-PKcs phosphorylates the RPA32 subunit at the diagnostic sites S4/8 (ref. 27). Thus, to confirm the end-resection-defective phenotype, we analysed formation of pS4/8-RPA32 foci after 4 h of CPT treatment; a period sufficient to allow formation of ssDNA, recruitment and phosphorylation of RPA, which is also crucial to the function of WRN in this pathway[28]. Consistently with the

ssDNA assay, detection of pS4/8-RPA32 foci showed a defective accumulation of positive nuclei in WS cells or in cells expressing the S1133A-WRN mutant (Fig. 2c). A similar result was observed when the presence of pS4/8-RPA32 was analysed by WB in chromatin fractions from CPT-treated cells (Supplementary Fig. 5D), and by visualization of RPA32 foci by immunofluorescence (Supplementary Fig. 5E).

Pharmacological inhibition of CDKs severely reduces end resection at DSBs[13]. Thus, to test whether the expression of the

phosphomimic S1133D-WRN could override the CDK-dependent regulation, we evaluated formation of ssDNA in WRN$^{S1133D}$ cells treated or not with roscovitine. As expected, roscovitine reduced the formation of ssDNA almost completely in wild-type and in WRN$^{S1133D}$ cells (Supplementary Fig. 6), suggesting that the presence of the phosphomimicking mutation is not sufficient to compensate for loss of activation of other end-processing factors by CDKs.

Depletion of CtIP or pharmacological inhibition of the MRE11 exonuclease by mirin also suppresses end resection at DSBs by interfering with the initial trimming step of the process[29,30]. Accordingly, CtIP RNAi (Supplementary Fig. 7A) or mirin treatment was sufficient to greatly reduce ssDNA formation already after 30 min of CPT exposure (Supplementary Fig. 7B,C). Of note, depletion of CtIP or treatment with mirin inhibited ssDNA formation also in cells expressing the S1133D-WRN mutant, confirming that mimicking constitutive phosphorylation by CDK1 is not sufficient to initiate resection and that WRN acts downstream CtIP and MRE11.

Therefore, these results demonstrate that phosphorylation of WRN by CDK1 is essential to promote the long-range step of the end resection, at least for replication-dependent DSBs. Furthermore, they suggest that phosphorylation of WRN by CDK1 occurs downstream of the initial resection by CtIP and MRE11.

**Phosphorylation of S1133 regulates the WRN-DNA2 pathway.** It was recently shown that WRN helicase activity stimulates DNA2 exonuclease in one of the two alternative long-range end-resection pathways[17]. To test if CDK1-mediated phosphorylation of WRN might affect the WRN-DNA2 pathway of end resection, we first examined formation of ssDNA by the IdU/ssDNA assay in WS cells complemented with a helicase-dead mutant of WRN (WRN$^{K577M}$). Consistently with previous reports[17], WRN$^{K577M}$ cells showed a significant reduction of the end resection at 90 min of treatment with CPT, as indicated by the greatly decreased ssDNA formation (Fig. 3a). Remarkably, the reduction in the ssDNA fluorescence intensity was comparable in cells expressing the helicase-dead (WRN$^{K577M}$) or the unphosphorylable WRN mutant (WRN$^{S1133A}$) (Fig. 3a). Moreover, pharmacological inhibition of WRN helicase activity by the small-molecule inhibitor NSC 617145 (ref. 31) failed to further reduce the formation of ssDNA in cells expressing the S1133A-WRN, but significantly decreased end resection in cells expressing the wild-type WRN or its S1133D mutant form (Supplementary Fig. 8).

As expression of S1133A or K577M WRN mutant similarly affected end resection, we wanted to determine if the unphosphorylable form of WRN retained its helicase and exonuclease activities. To this end, we purified recombinant wild-type WRN or the two phosphorylation mutants (S1133A and S1133D; Supplementary Fig. 9), and analysed their enzymatic activities on a forked duplex DNA substrate that is unwound by WRN in the presence of ATP or degraded by WRN's 3′–5′ exonuclease activity in the absence of ATP[32]. As shown in Fig. 3b,c, no apparent difference in the helicase or exonuclease activity was detected between the unphosphorylable WRN mutant and the wild-type protein, and, most interestingly, the enzymatic activities of WRN were also unchanged by the presence of the phosphomimetic S1133D substitution.

Although the DNA2-dependent long-range end resection is strongly stimulated by WRN *in vitro*, depletion of WRN alone is not sufficient to block long-range end resection *in vivo*, as BLM may compensate for the absence of WRN[17]. To verify if the residual ssDNA detected at 90 min post CPT treatment in WRN$^{S1133A}$ cells might derive from WRN-independent, DNA2-dependent resection, we analysed formation of ssDNA in cells depleted of DNA2 by RNAi (Fig. 4a). Remarkably, depletion of DNA2 significantly reduced the formation of ssDNA in WS cells and in cells expressing WT-WRN or S1133D-WRN (Fig. 4a), confirming that a DNA2-dependent end resection takes place even in the absence of WRN[17]. In sharp contrast, DNA2 depletion was completely ineffective in further reducing ssDNA formation (that is, end resection) in cells expressing S1133A-WRN (Fig. 4a).

In the absence of WRN-DNA2, long-range end resection can be performed by EXO1, alone or in combination with BLM[16]. To investigate the genetic dependency of the residual ssDNA observed in the WRN phosphomutants, we performed RNAi to deplete EXO1 and analysed end resection by the IdU/ssDNA assay at 90 min post CPT treatment. As shown in Fig. 4b, depletion of EXO1 only marginally reduced ssDNA formation in wild-type cells and failed to inhibit further the level of ssDNA in cells expressing the S1133A-WRN mutant. Notably, formation of ssDNA in the S1133D-WRN mutant was only marginally affected by EXO1 depletion, whereas it was significantly reduced in WS cells. As expression of the S1133A-WRN mutant appeared to prevent engagement of other end-processing pathways, we analysed formation of BLM foci by immunofluorescence after CPT treatment in WS cells complemented with the wild-type WRN or the S1133A mutant (Supplementary Fig. 10). BLM foci-positive nuclei were found to accumulate in the parental WS cells threefold more than in cells expressing the wild-type WRN or the S1133-WRN (Supplementary Fig. 10).

Altogether, these results demonstrate that phosphorylation of S1133 is instrumental to the WRN helicase-DNA2-dependent long-range resection pathway. They also suggest that at least for replication-dependent DSBs, loss of CDK1-dependent phosphorylation of WRN abrogates almost completely long-range end resection, interfering with compensatory pathways, whereas constitutive phosphorylation of WRN at S1133 strongly selects for the WRN-DNA2-dependent pathway.

**Phosphorylation at S1133 promotes replication fork restart.** A striking characteristic of WS cells is the reduced replication fork rate[33,34] and the weakened replication recovery after CPT treatment[35,36]. Therefore, we investigated whether CDK1-dependent phosphorylation of WRN was involved in replication fork progression under unperturbed cell growth or during recovery from CPT treatment. To this aim, we performed single-molecule analysis of DNA replication using the DNA fibre technique after dual-labelling with CldU, to mark ongoing forks, and then with IdU to quantify their progression in the presence or not of DNA damage (Fig. 5a). As expected, WS cells showed a reduced replication fork progression compared with cells complemented with the wild-type WRN, as evaluated by the shorter IdU tracks (Fig. 5b). Expression of S1133A-WRN was sufficient to recover this DNA replication defect almost completely while introduction of S1133D-WRN completely reverted the phenotype (Fig. 5b). Next we examined the ability of the WRN phosphorylation mutants to recover DNA replication after CPT treatment (Fig. 5c). In the absence of WRN, a 35% reduction of the fork restart after CPT treatment was observed (Fig. 5c). Defective replication fork recovery was reverted by complementation with wild-type WRN or the S1133D-WRN but not by expression of the unphosphorylable WRN mutant (Fig. 5c).

These findings indicate that loss of the CDK1-regulated function of WRN is unrelated to reduced replication fork rate characteristic of WS cells, however, CDK1-dependent phosphorylation of WRN is essential to recover replication following fork collapse.

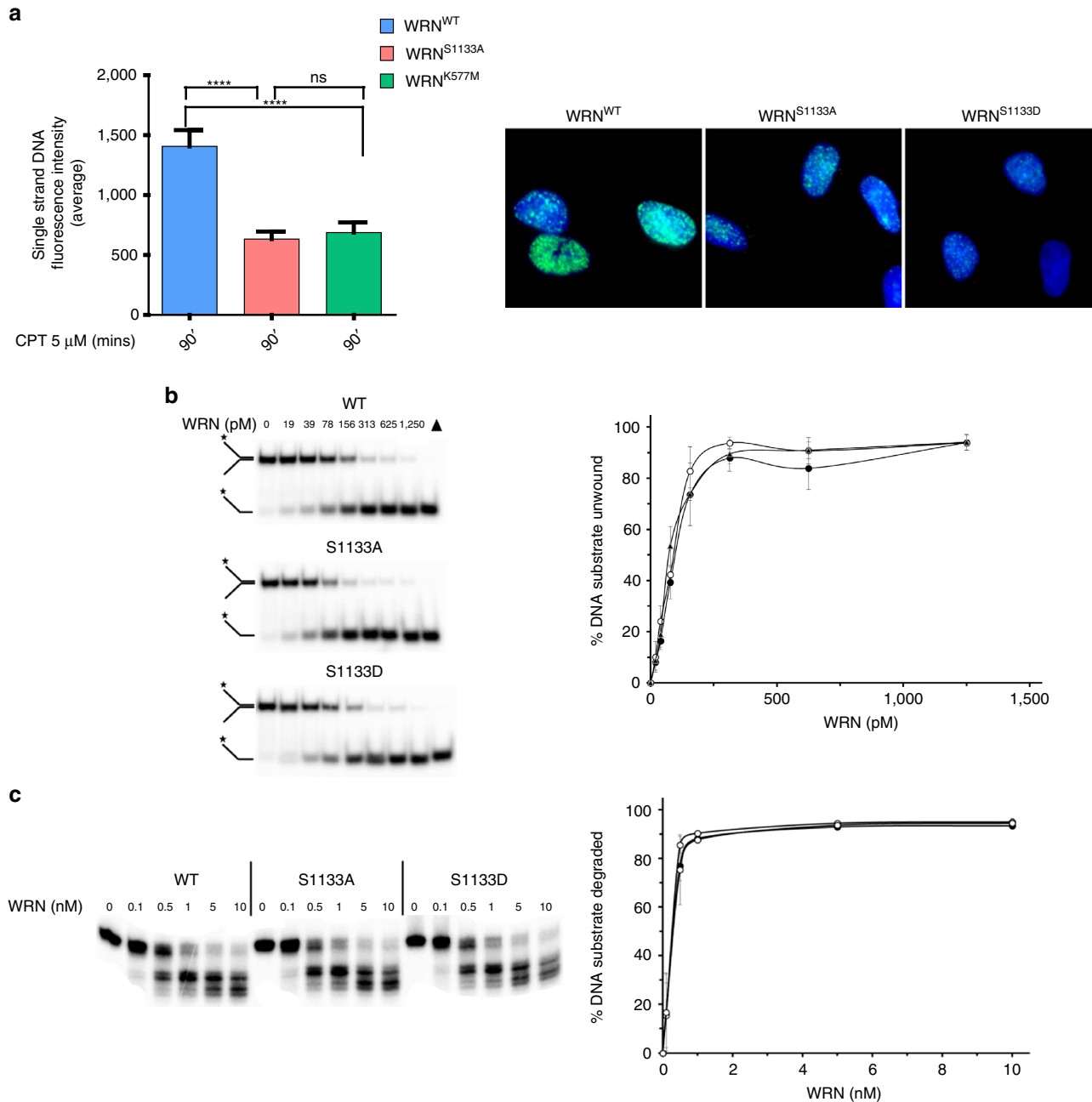

**Figure 3 | CDK1-dependent phosphorylation of WRN at S1133 affects the WRN helicase-DNA2-dependent pathway of end resection.** (**a**) WS-derived cells stably expressing the WRN wild type (WRN^WT), the unphosphorylable (WRN^S1133A) or the helicase-dead (WRN^K577M) mutant were labelled and treated as in Fig. 2b, and end resection analysed by the non-denaturing IdU/ssDNA assay. The graph shows the mean intensity of IdU/ssDNA staining. The intensity of the anti-IdU immunofluorescence was measured from two independent experiments ($n = 100$, each biological replicate), data are presented as mean ± s.e.m. The panel shows representative images of IdU/ssDNA detection after CPT treatment. (**b**) Flag-WRNWT, Flag-WRNS1133A and Flag-WRNS1133D have comparable helicase activity on a forked DNA substrate. A unit of 0–1,250 pM Flag-WRNWT, Flag-WRNS1133A and Flag-WRNS1133D were incubated with forked DNA substrate as described in the Methods section. Quantitation of panels. Data represents the average of a minimum of three independent experiments. (**c**) Flag-WRNWT, Flag-WRNS1133A and Flag-WRNS1133D have comparable exonuclease activity on a forked DNA substrate. A unit of 0–10 nM Flag-WRNWT, Flag-WRNS1133A and Flag-WRNS1133D were incubated with forked DNA substrate as described in the Methods section. Quantitation of **a**. Data represent the average of a minimum of three independent experiments.

**CDK1 regulates association of WRN with MRE11.** On CPT treatment, WRN assembles in nuclear foci[37,38]. To determine whether phosphorylation at S1133 could affect recruitment of WRN at collapsed forks, we analysed formation of WRN nuclear foci in cells expressing the two WRN phosphomutants. Mutation abrogating CDK1-dependent phosphorylation or mimicking constitutive phosphorylation at S1133 did not affect ability of WRN to localize at nucleoli under unperturbed cell growth (Supplementary Fig. 11) or in nuclear foci after CPT (Fig. 6a). The sub-nuclear relocalization of WRN at collapsed forks is greatly reduced in the absence of MRE11 (ref. 22). Thus, we investigated whether phosphorylation of S1133 could be involved in the MRE11-dependent WRN relocalization. To this end, we depleted MRE11 by RNAi in cells expressing the wild-type form

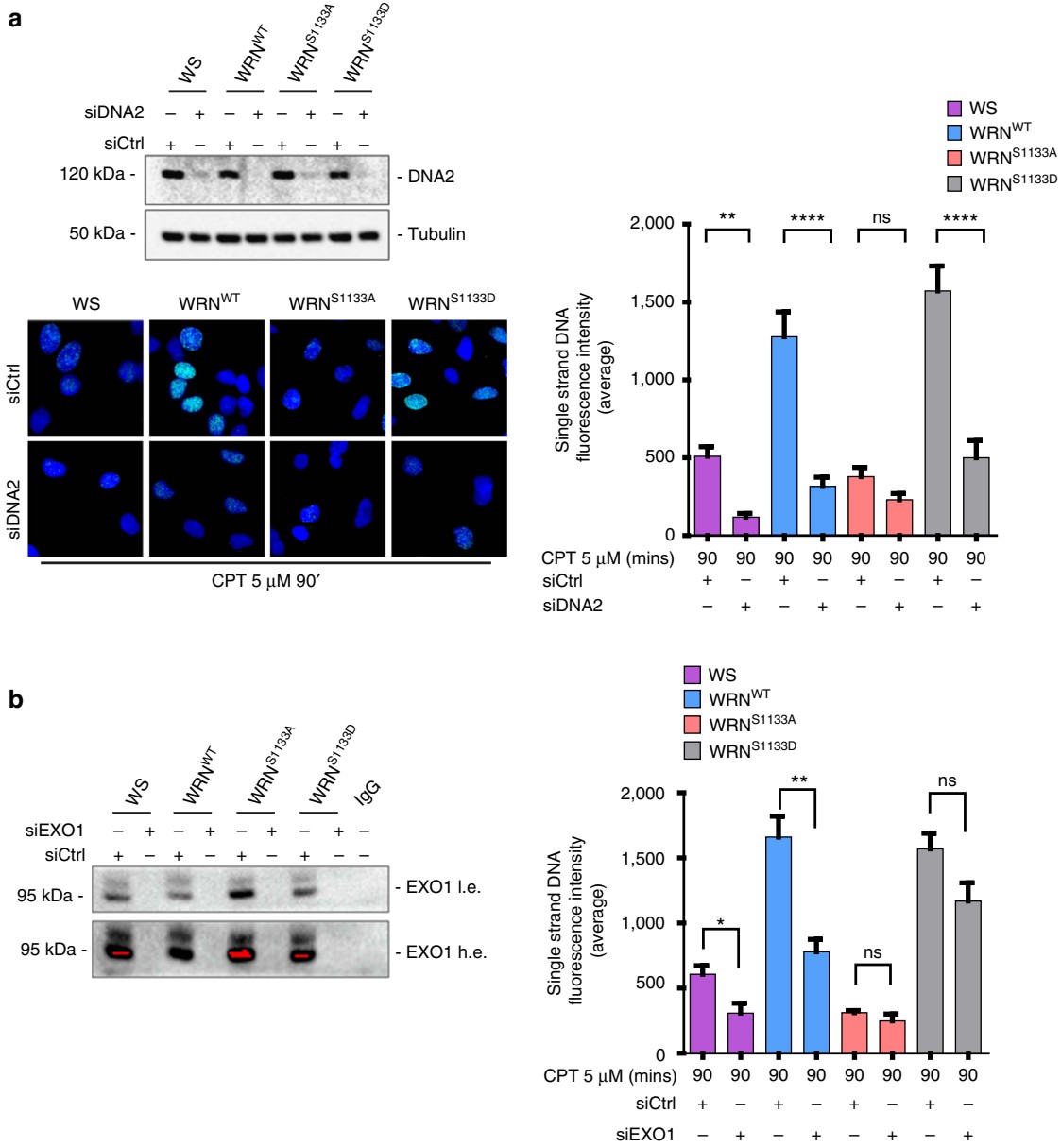

**Figure 4 | CDK1-dependent phosphorylation of WRN at S1133 affects the WRN helicase-DNA2-dependent pathway of end resection. (a)** Loss of WRN phosphorylation impinges on DNA2-dependent end resection. Western blotting shows depletion of DNA2 in cells expressing WRN wild type or its phosphorylation mutants. Whole-cell extracts were prepared at 48 h after transfection with DNA2 siRNA. Representative images of control (Ctrl)-depleted or DNA2-depleted cells treated with CPT and immunostained for IdU/ssDNA are shown. The graph shows the mean intensity of IdU/ssDNA staining in cells treated or not with the DNA2 siRNA. The intensity of the anti-IdU immunofluorescence was measured in two independent experiments ($n=100$, each biological replicate), data are presented as mean ± s.e.m. **(b)** Loss of WRN S1133 phosphorylation affects EXO1-mediated degradation. Western blotting shows depletion of EXO1 in cells expressing WRN wild type or its phosphorylation mutants, or no WRN (WS). Whole-cell extracts were prepared at 48 h after transfection with EXO1 siRNA and immunoprecipitated with an anti-EXO1 antibody, followed by WB with the same antibody. End resection was evaluated by the IdU/ssDNA assay. The graph shows the mean intensity of ssDNA staining for single nuclei quantitated from two independent experiments ($n=100$, each biological replicate), data are presented as mean ± s.e.m. Statistical analysis was performed by the analysis of variance test (****$P<0.0001$; **$P<0.01$; *$P<0.05$; NS, not significant).

of WRN or the two WRN phosphorylation mutants (Fig. 6b) and analysed formation of WRN nuclear foci after treatment with CPT. As expected, depletion of MRE11 severely compromised the formation of WRN foci, as demonstrated by the significant reduction of the intensity of WRN nuclear-foci fluorescence (Fig. 6c). In contrast, depletion of MRE11 did not affect accumulation of WRN in nuclear foci either in the presence of the unphosphorylable S1133A or of the phosphomimetic S1133D mutation (Fig. 6c). Thus, we asked whether S1133

phosphorylation could be involved in the association of WRN with the MRE11 complex after replication-induced DSBs. Accordingly, we performed *in situ* proximity-ligation assay (PLA)[39] to detect physical association of WRN and the MRE11 complex after treatment. While no PLA signal was detectable under unperturbed conditions (Supplementary Fig. 12), an elevated fraction of PLA-positive cells was readily seen in cells expressing wild-type WRN on CPT treatment (Fig. 6d,e). Similarly, CPT treatment resulted in the appearance of WRN-

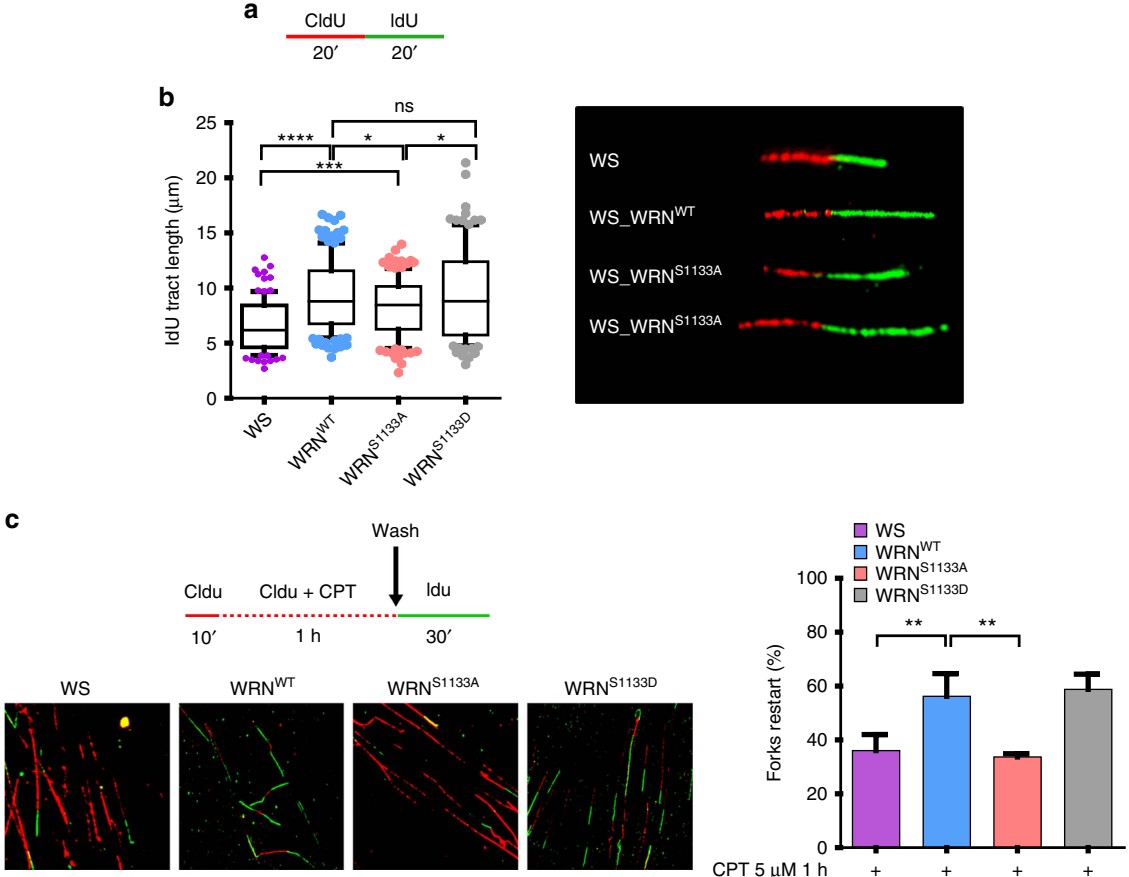

**Figure 5 | Replication restart after CPT-mediated fork collapse is affected by loss of WRN phosphorylation at S1133.** (**a**) Labelling scheme used to analyse replication fork progression under unperturbed conditions by DNA fibre assay. (**b**) Box and whiskers graphs showing the 10–90 percentile of the IdU tract length (μm) for ongoing forks. Length of the IdU-labelled green tracts was measured in at least 100 well-isolated DNA fibres from two independent experiments ($n = 100$). Data are presented as mean ± s.e.m. (NS, not significant; *$P < 0.1$; ***$P < 0.001$; ****$P < 0.0001$; Mann–Whitney test). (**c**) Analysis of replication fork restart in CPT-treated cells. Cells were labelled and treated as indicated in the experimental scheme to estimate the number of restarting forks after DNA damage. The graph shows the percentage of restarting forks calculated on the total number of forks, excluding that fired during recovery (IdU-labelled only). At least 150 single replication forks were analysed for each point from two independent experiments and data are presented as mean ± s.e.m. (**$P < 0.01$; analysis of variance test; $n = 150$). Representative images of DNA fibres from the different cell lines are shown in the panel.

MRE11 PLA spots also in cells expressing S1133D-WRN, in which, however, the percentage of PLA-positive cells and the number of PLA spots were reduced as compared with the wild type (Fig. 6d,e). The percentage of nuclei showing interaction between the MRE11 complex and WRN, as well as the number of PLA spots, was also decreased in WRN^S1133A cells (Fig. 6d,e). Of note, loss of S1133 phosphorylation of WRN decreased the number of MRE11-positive nuclei at later time points after treatment (2 and 3 h), although the number of MRE11 foci per nucleus was not greatly affected (Supplementary Fig. 13). As such, the defective interaction between S1133A-WRN and the MRE11 complex is not just a secondary effect of the reduced number of MRE11-positive cells, since expression of S1133A-WRN greatly decreased also the number of WRN-MRE11 PLA spots/nucleus. Surprisingly, expression of S1133D-WRN resulted in a large reduction in both the percentage of MRE11-positive nuclei and the number of MRE11 foci at 3 h of treatment (Supplementary Fig. 13).

These results indicate that phosphorylation of WRN by CDK1 is essential for the correct formation of MRE11 foci and the WRN interaction with the MRE11 complex. Moreover, they suggest that a mutation mimicking constitutive phosphorylation of WRN

dissociates its recruitment from the presence of MRE11 and leads to disassembly of MRE11 foci.

**WRN phosphorylation regulates DSB repair pathway choice.** As CDK1-mediated modification of WRN is critical to promote end resection, we reasoned that such a crucial modification could be relevant for repair pathway choice. To address this question, we used two different reporter assays to test the efficiency of endonuclease-induced DSB repair by HR or NHEJ[40,41]. The pDRGFP HR reporter was transiently transfected in HEK293TshWRN cells together with a plasmid expressing the wild type, S1133A or S1133D variants of WRN and a plasmid expressing the I-SceI endonuclease. At 72 h post transfection, repair was evaluated by flow cytometry analysing the number of GFP-positive cells[41]. As shown in Fig. 7a, the efficiency of HR was slightly reduced in cells expressing the S1133A-WRN, while it was increased more than twofold by the presence of the WRN-S1133D phosphomimetic mutation. In contrast, expression of the unphosphorylable WRN-S1133A enhanced the efficiency of NHEJ that was not significantly affected by the presence of the S1133D-WRN mutant (Fig. 7b). Consistent with enhanced HR,

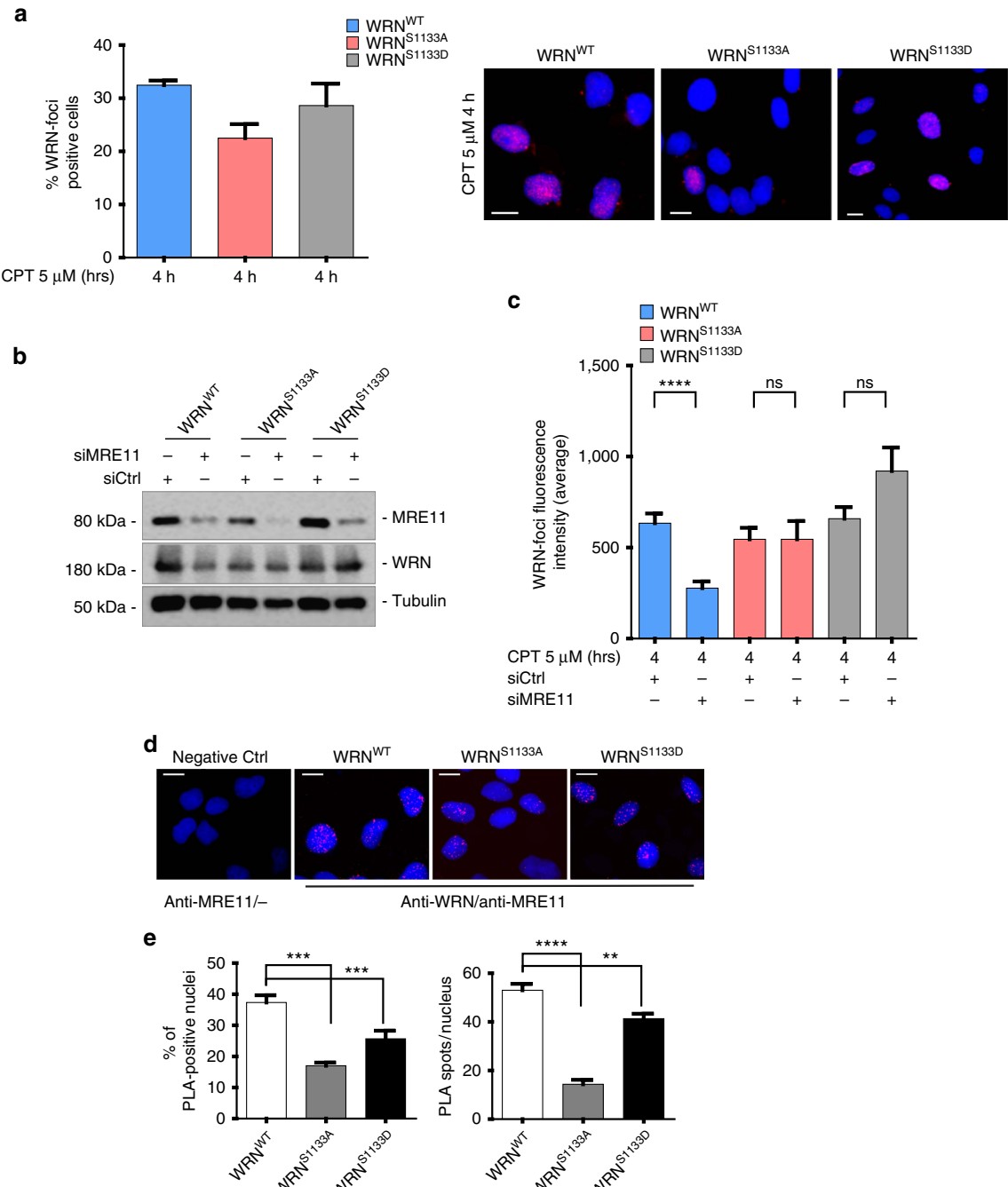

**Figure 6 | Phosphorylation at S1133 is involved in regulating the association of WRN with the MRE11 complex and MRE11 foci-formation after replication-dependent DSBs.** (**a**) Analysis of WRN foci formation by immunofluorescence. Cells were treated with CPT as indicated and then analysed for the presence of WRN foci by immunofluorescence with an anti-WRN antibody. The graph shows the percentage of WRN-foci positive cells counted in random fields. For each time point, at least 200 cells were analysed from three independent experiments. Data are presented as mean ± s.e.m. ($n = 300$). The panel shows representative images of WRN foci formation in response to treatment. DAPI was used to counterstain nuclei. Scale bar, 5 μm. (**b**) Western blotting shows MRE11 depletion. Whole-cell extracts were prepared 48 h after transfection with MRE11or control (Ctrl) siRNA. (**c**) Cells were transfected with Ctrl or MRE11 siRNAs, and then analysed for the formation of WRN foci after CPT treatment, as indicated. The graph shows the quantification of the mean fluorescence intensity of WRN foci-positive cells from random fields. For each time point, at least 200 cells were analysed from three independent experiments. Data are presented as mean ± s.e.m. (****$P < 0.0001$; NS, not significant, analysis of variance (ANOVA) test; $n = 200$). (**d**) Analysis of WRN-MRE11 interaction by *in situ* PLA. Cells were treated with 5 μM CPT for 2 h and subjected to PLA using anti-WRN and anti-MRE11 antibodies. The panel shows representative PLA images showing association of WRN with MRE11. The negative control shows the results of PLA with only the anti-MRE11 antibody. (**e**) The graphs show the number of PLA-positive nuclei (right) and that of PLA spots per PLA-positive cell (left). At least 300 nuclei were analysed for each experimental point from two independent experiments. Values are presented as mean ± s.e. (**$P < 0.01$; ***$P < 0.001$; ****$P < 0.0001$; ANOVA test; $n = 300$). Scale bar, 10 μm.

recruitment of RAD51 in chromatin was increased after 6 h of treatment in wild-type cells and, even more, in cells expressing the WRN-S1133D mutant, while it was barely affected in cells expressing the unphosphorylable WRN mutant (Supplementary Fig. 14). Moreover, expression of the S1133D-WRN mutant also

resulted in a threefold increase of sister chromatid exchanges (SCEs) (Fig. 7c), a cytological sign of elevated recombination.

To evaluate whether the apparent pathway switch observed in WRN$^{S1133A}$ cells affected the rate of DSB repair, we measured the extinction of DNA damage at different time points of recovery

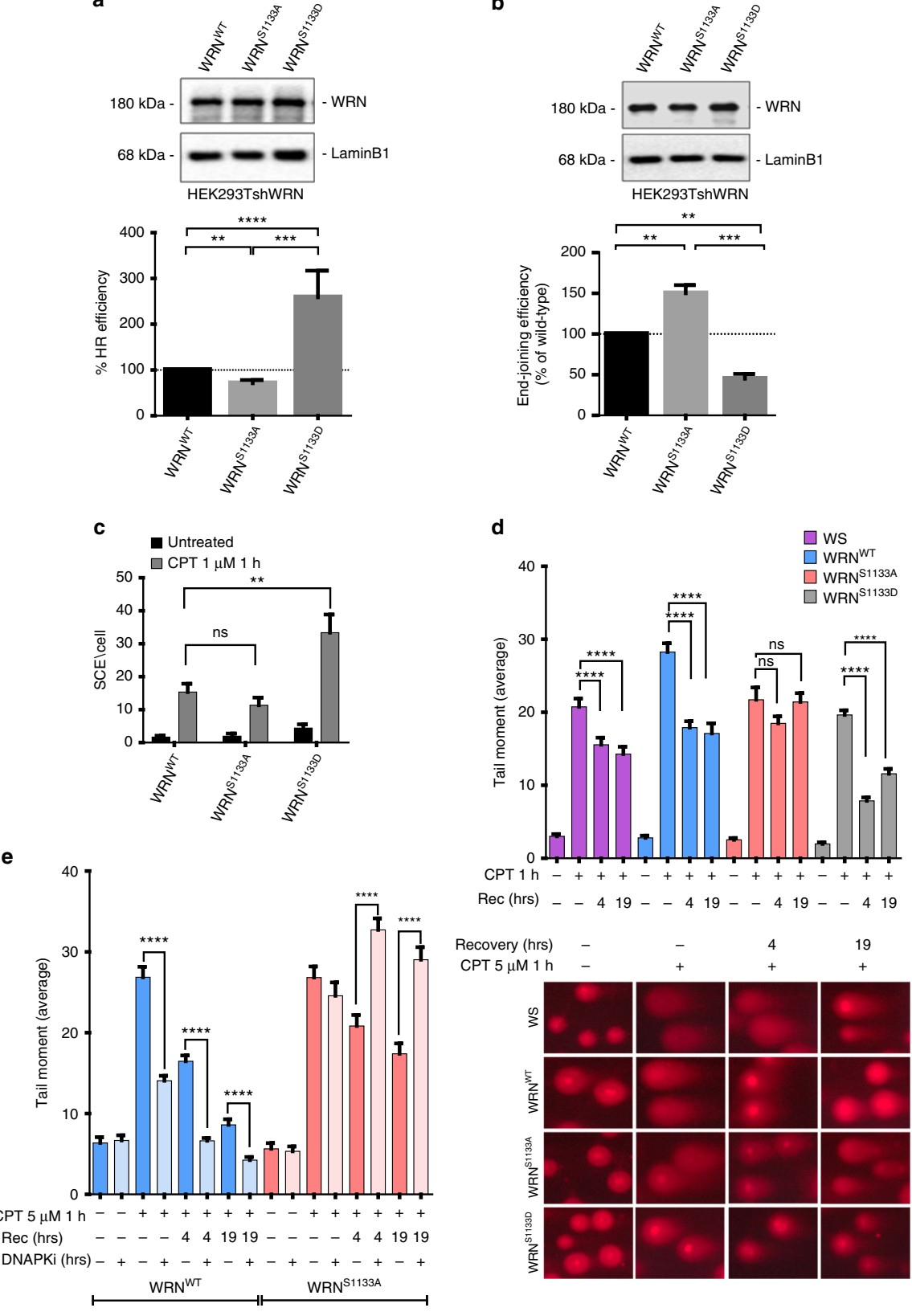

after CPT treatment using the neutral comet assay. In wild-type cells, the DSBs induced by CPT dropped to about 70% of the starting level after recovery, while they reduced of only 10% in cells expressing the S1133A-WRN (Fig. 7d). In contrast, expression of S1133D-WRN significantly improved repair of DSBs after CPT exposure (Fig. 7d). Interestingly, the ability of the parental WS cells to repair DSBs appeared intermediate between the wild-type and the WRN[S1133A] cells (Fig. 7d).

Reduced HR repair observed in WS cells expressing the S1133A-WRN mutant and the concomitant elevated NHEJ usage detected with the reporter assays, prompted us to verify whether residual repair of CPT-induced DSBs was attributable to NHEJ. We reasoned that repair of DSBs, observed as decrease of the tail moment value, would be reduced after inhibition of NHEJ much more in WRN[S1133A] cells than in wild-type or S1133D-WRN expressing cells. As shown in Fig. 7e, the DNA-PKcs inhibitor NU7441 did not reduce, but apparently ameliorated, DSB repair in wild-type cells or in cells expressing the phosphomimetic WRN mutant. In contrast, DNA-PKcs inhibition substantially abrogated repair in cells expressing S1133A-WRN. These results suggest that phosphorylation of WRN by CDK1 may regulate the balance of the DSB repair pathways. Next, we examined whether the inability to phosphorylate WRN could undermine genome integrity and cell viability. Genome integrity was evaluated on metaphase spreads after recovery from CPT treatment in WS-derived cells complemented with the wild-type form of WRN or the two phosphomutants. As expected, WRN deficiency resulted in more chromosomal breaks and exchanges compared with the wild-type cells (Fig. 8a,b). Strikingly, the number of chromosome exchanges (radial chromosomes) was not different in wild-type and S1133D-WRN cells, while it was significantly enhanced in cells expressing the S1133A-WRN mutant (Fig. 8b). Surprisingly, and in spite of the apparent elevated HR repair efficiency, the cells expressing the phosphomimetic WRN mutant (WRN[S1133D]) were the most sensitive to the CPT treatment as evaluated by clonogenic survival or a short-term live/dead viability assay (Fig. 8c and Supplementary Fig. 15).

The elevated HR repair associated with expression of the phosphomimetic WRN mutant along with the reduced viability observed after treatment in these cells, prompted us to test whether cell death could derive from hyper-recombination. To this end, we compared viability of cells expressing the WRN phosphomutants in the presence or not of B02, a RAD51 chemical inhibitor[42]. Interestingly, while inhibition of RAD51 by itself increased cell death in the absence of CPT treatment irrespective of the presence of wild-type WRN, it largely rescued cell death observed after CPT-induced DSBs in WRN[S1133D] cells (Fig. 8d). Consistently, cell death of WRN[S1133D] cells was also

reduced by over-expression of a dominant-negative RAD51 chimera that interferes with nucleofilament formation[43] (smRAD51; Supplementary Fig. 16A,B). Since inhibition of RAD51 did not alter sensitivity of WRN[S1133A] cells to CPT (Fig. 8d), and given that loss of CDK-dependent WRN regulation seems to favour NHEJ, we tested whether the reduced sensitivity of WRN[S1133A] cells to CPT was correlated to NHEJ activation. Thus, we treated cells with CPT and analysed viability in the presence or absence of NU7441. Inhibition of DNA-PKcs significantly elevated cell death of cells expressing the S1133A-WRN protein but did not affect viability of wild-type or WRN[S1133D] cells after CPT treatment (Supplementary Fig. 16C).

Altogether, these findings indicate that phosphorylation of WRN by CDK1 is required to channel replication-dependent DSBs through the HR pathway. They also show that abrogation of phosphorylation by CDK1 at S1133 of WRN hampers HR, resulting in elevated NHEJ-mediated repair of DSBs at collapsed forks and chromosome instability.

## Discussion

Regulation of DSB repair is crucial for maintenance of genome integrity in mammalian cells. Here we describe a novel regulatory mechanism controlling long-range resection at replication-dependent DSBs and repair pathway choice through the phosphorylation of WRN mediated by CDK1.

Similar to yeast, human CDKs regulate end resection and pathway choice mainly controlling the function of key proteins involved in the initial steps of end processing, such as CtIP or NBS1 (refs 8,13,44).

Much less is known about the regulation of long-range end resection in human cells, and only recently has it been shown that the activity of EXO1 is regulated by CDKs[14]. However, in human cells, long-range end resection can be performed also by DNA2 together with the WRN or BLM helicase[16,17], but very little is known about its regulation. In this study, we find that CDK1 phosphorylates the C-terminal region of WRN at S1133 influencing processing of DSBs at the replication fork. Phosphorylation of WRN by CDK1 is not required for the initial processing of the DSB, but is crucial to sustain the subsequent long-range end resection, corroborating previous evidence for a role of WRN at this stage[17,45,46]. Interestingly, phosphorylation of WRN at S1133 influences the DNA2-dependent resection pathway, and thus provides a novel regulatory level to the long-range end resection. In human cells, depletion of either RecQ helicase fails to completely suppress end resection[17]. Our data show that expression of S1133A-WRN or DNA2 depletion similarly impairs ssDNA formation at

**Figure 7 | CDK-dependent phosphorylation of WRN at S1133 regulates DSB repair pathway choice at collapsed forks.** (**a**) Efficiency of HR-mediated repair of I-SceI-induced DSBs in HEK293TshWRN cells co-transfected with the indicated WRN forms, the I-SceI expression vector pCBASce and the pDRGFP HR reporter plasmid, as described in Methods. Western blotting shows expression of the indicated WRN forms in the HEK293TshWRN cells. The graphs show the percentage of HR efficiency calculated respect to cells transfected with the wild-type WRN protein. (**b**) End-joining efficiency of I-SceI-induced DSBs in HEK293TshWRN cells co-transfected with the indicated WRN forms, the pDRGFP NHEJ reporter plasmid linearized, as described in Methods. Western blotting shows expression of the indicated WRN forms in the HEK293TshWRN cells. The graph shows the percentage of NHEJ efficiency respect to cells transfected with the wild-type WRN. Data are presented as mean ± s.e.m. from three independent experiments (**P < 0.01; ***P < 0.001; ****P < 0.0001; analysis of variance (ANOVA) test; $n = 3 \times 10^5$). (**c**) Analysis of sister chromatid exchanges. Cells were treated as indicated and recovered in BrdU-containing medium for 36 h before metaphase spreading and staining, as described in Supplementary Methods. The graph shows the mean number of sister chromatid exchanges (SCE) per metaphase cells. A minimum of 25 metaphases were counted for each experimental point from three independent experiments (NS, not significant; **P < 0.01, ANOVA test; $n = 75$). (**d**) Analysis of DSB repair efficiency. Cells were treated with 5 µM CPT for 1 h and allowed to recover for different time points as indicated. DSB repair was evaluated by the neutral Comet assay. In the graph, data are presented as mean tail moment ± s.e.m. from three independent experiments (NS, not significant; ****P < 0.0001; Mann–Whitney test; $n = 300$). Representative images from the neutral Comet assay are shown in the panel. (**e**) Effect of DNA-PKcs inhibition on DSBs repair in the WRN phosphomutants. Cells were treated as indicated and allowed to recover in the presence or not of the DNA-PKcs inhibitor (DNA-PKi). The presence of

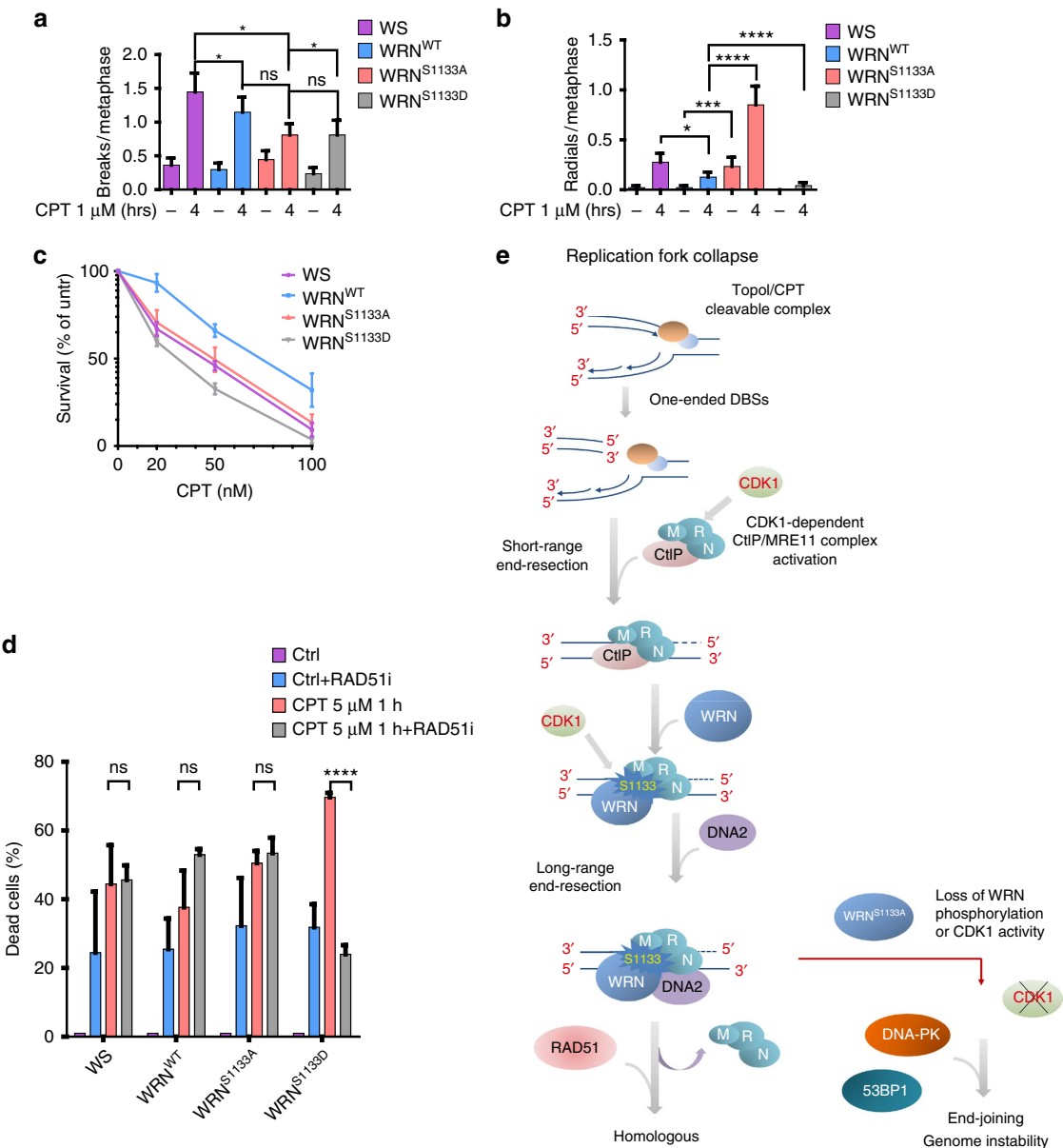

**Figure 8 | CDK-dependent phosphorylation of WRN is required for genome integrity and survival in response to replication-dependent DSBs.**
(**a**,**b**) Graphs show the average number per cell of the indicated chromosomal aberrations in WS and the different WS-complemented cells. Cells were treated with 1 μM CPT 4 h followed by 24 h of recovery in drug-free medium before collecting. Data are presented as mean ± s.e.m. of three independent experiments (*$P < 0.05$; ***$P < 0.001$; ****$P < 0,0001$; NS, not significant, analysis of variance (ANOVA) test; $n = 150$). (**c**) Analysis of clonogenic survival. Cells expressing the wild-type or the two phosphomutant form of WRN were treated with CPT for 24 h at the indicated doses and the number of colonies for each experimental point was recorded and expressed as a percentage of the untreated control. Data are mean ± s.e.m. from two independent experiments. (**d**) RAD51 inhibition in WRN phosphomimetic mutant expressing cells modulates sensitivity to CPT. Cells were treated with CPT for 1h, as indicated. The B01 RAD51 inhibitor was added before CPT treatment, and then cells were recovered for 18 h in the continuous presence or not of the inhibitor before assessing cell death by LIVE/DEAD assay. Data are presented as percentage of cell death and are mean values from three independent experiments (****$P < 0.0001$; ***$P < 0.001$, ANOVA test; NS, not significant; $n = 3$). (**e**) Proposed model of CDK-dependent regulation of WRN in the repair of DSBs at the replication fork (see text for details).

CPT-induced DSBs, suggesting that the CDK1-unphosphorylable WRN behaves as a dominant-negative mutant, precluding BLM from taking over for WRN[17]. Although the long-range end resection can be performed by EXO1 when the DNA2 pathway is not functional[45], we observe that depletion of DNA2, expression of the S1133A-WRN mutant or inactivation of the WRN helicase has much more profound effect than depletion of EXO1 on ssDNA formation after CPT treatment. Much of the results indicating that the EXO1 and the DNA2 pathways are equivalent,

and almost interchangeable, were derived from processing of two-ended DSBs. Our data suggest that, at least after replication-dependent, one-ended, DSBs induced by CPT, the role of WRN-DNA2 predominates over that of EXO1. Interestingly, the presence at the break of the CDK-unphosphorylable WRN prevents effective engagement of any other long-range resection mechanism, as suggested by the phenotype of WRN[S1133A] cells depleted of EXO1. From this point of view, CDK1-dependent regulation of WRN is reminiscent of the Cdk-dependent

regulation of Dna2 in yeast[47]. Further studies will be required to define whether one-ended DSBs are differentially handled from canonical two-ended DSBs.

WRN and the MRE11 complex interact in response to DSBs or fork stalling[22,48]. We show that loss of CDK1-dependent phosphorylation affects interaction of WRN with MRE11 and decreases the number of MRE11-positive nuclei after CPT. It has been proposed that the MRE11 complex promotes long-range end resection regulating the loading and function of EXO1 and DNA2 (refs 16,49). Consistently, inability of the S1133A-WRN to support long-range resection in the DNA2-mediated pathway and the dominant-negative effect on that EXO1-dependent may be linked to defective formation of MRE11 on DSBs. Interestingly, although the S1133A-WRN retains a wild-type ability to relocalize in nuclear foci after DSBs, this occurs independently of the presence of MRE11, which is otherwise essential in wild-type cells[22]. From this point of view, the altered association of S1133-WRN with MRE11 may correlate with engagement of a different set of proteins at collapsed forks. Indeed, expression of the S1133A-WRN causes a switch from HR to NHEJ, as shown by reduced HR efficiency, loss of DNA damage-induced RAD51 chromatin binding and increased dependency to DNA-PKcs for repair and viability in response to CPT. It has been shown that WRN participates also in NHEJ and is phosphorylated by DNA-PKcs[50]. Thus, WRN is an ideal candidate to control a DSB repair pathway switch, and this role would be consistent with the observed dominant-negative effect of S1133A-WRN. Of note, radial chromosomes are greatly increased in WRN[S1133A] cells after CPT treatment, more than in the parental WS cells in which the total amount of chromosomal breaks is higher. Since WRN-deficient cells are still partially proficient for the long-range resection[17], distinct pathways may be responsible for chromosome instability in the absence of WRN[51].

Consistently with a key role of WRN phosphorylation in a DSB repair pathway switch, S1133D-WRN strongly enhances frequency of HR events in the reporter assay, and also increases the recruitment of RAD51 in chromatin after CPT. Interestingly, S1133D-WRN also bypasses the requirement of MRE11 for recruitment in foci and, surprisingly, its expression leads to a faster dismantling of MRE11 foci after CPT consistently with the enhanced loading of RAD51 in chromatin. Thus, a mutation mimicking constitutive CDK1 phosphorylation may overcome any regulated WRN association at DSBs and DNA2-WRN helicase-dependent long-range resection, consistent with the apparent faster DSBs repair and hyper-rec phenotype. Moreover, S1133D-WRN sensitizes cells to CPT in a RAD51-dependent manner, suggesting that toxic recombination intermediates accumulate in WRN[S1133D] cells.

WRN plays crucial roles in the recovery from perturbed replication that are unrelated to HR[20,34]. Loss of CDK1-dependent regulation of WRN in the end resection is essential for the correct recovery of collapsed replication forks, but it is irrelevant for the unperturbed replication. Indeed, S1133A-WRN is able to revert almost completely the reduced replication fork progression characteristic of WS cells[26,52,53]. This would indicate that the role of WRN in DSB end resection and the replicative WS phenotype may be largely unrelated, in sharp contrast to what was observed for ATR-dependent phosphorylation[20], and emphasizes the importance of investigating the regulatory network of WRN also to unveil the disease pathogenesis. Interestingly, a single-nucleotide polymorphism resulting in the S1133A change and several cancer somatic mutations involving the region surrounding S1133 have been recorded in the COSMIC database, suggesting that loss of CDK1-dependent WRN regulation may be relevant for genome instability in cancer cells or as a predisposing factor in the population.

Therefore, our data and published observations can be summarized in a model (Fig. 8e), whereby CDK1 phosphorylation at S1133 supports interaction with the MRE11 complex, contributing to stabilize the MRE11 complex and recruiting DNA2 for the WRN-DNA2-dependent long-range resection. Inability to phosphorylate S1133 of WRN, as it occurs in G1 cells because of low CDK1 activity, would stimulate MRE11 dissociation, block further end resection and promote DNA-PKcs-mediated NHEJ at CPT-induced DSBs.

Thus, our study identifies a novel regulatory layer of WRN that specifically affects its function during HR and in particular during end resection, unveiling a previously uncharacterized function of WRN as a switch in DSB repair pathway choice. Loss of CDK1-regulated WRN function would lead either to unscheduled NHEJ or excessive HR, thus contributing to genome instability or cell death.

## Methods

**Cell lines and culture conditions.** The SV40-transformed WS cell line (AG11395) was obtained from Coriell Cell Repositories (Camden, NJ, USA). The AG11395 cells carries an Arg368 stop mutation that results in a truncated protein that is degraded and undetectable. To produce stable cell lines, AG11395 (WS) fibroblasts were transduced with lentiviruses expressing the full-length cDNA encoding wild-type WRN (WRN[WT]), S1133A-WRN (WRN[S1133A]) or S1133D-WRN (WRN[S1133D]). The WRNWT and the AG11395 complemented with a helicase-dead WRN (WRN[K577M]) have been described elsewhere[54]. HEK293T cells were from American Type Culture Collection. HEK293TshWRN, cells were generated after transfection with pRS-puro-shWRN (5′-AGGCAGGTGTAG- GAATTGA AGGAGATCAG-3′; sequence ID: TI333414 Origene) and puromicin selection. All the cell lines were maintained in Dulbecco's modified Eagle's medium (DMEM; Life Technologies) supplemented with 10% FBS (Boehringer Mannheim) and incubated at 37 °C in a humidified 5% $CO_2$ atmosphere. All cell lines have been tested for the presence of mycoplasma by PCR and DAPI-IF.

**Chemicals.** CPT (ENZO Lifesciences) was dissolved in dimethylsulfoxide, and a stock solution (10 mM) was prepared and stored at − 20 °C. CDK1 inhibitor RO-3306 (Selleck) was used at final concentration of 9 µM. Mirin (Calbiochem), an inhibitor of MRE11 exonuclease activity, was used at 50 µM; the B02 compound (Selleck), an inhibitor of RAD51 activity, was used at 27 µM. Roscovitine (Selleck), a pan-CDKs inhibitor, was used at final concentration of 20 µM. NU7441 (Selleck), a DNAPKcs inhibitor, was used at final concentration of 1 µM. The WRN helicase inhibitor (NSC 617145) was used at final concentration of 750 nM (ref. 31). IdU and CldU (Sigma-Aldrich) were dissolved in sterile DMEM at 2.5 and 200 mM, respectively, and stored at − 20 °C.

**Immunoprecipitation and western blot analysis.** Immunoprecipitation experiments are performed using $2.5 \times 10^6$ cells. RIPA buffer (0.1% SDS, 0.5% Na-dehoxycolate, 1% NP40, 150 mM NaCl, 1 mM EDTA, 50 mM Tris/Cl (pH 8)) supplemented with phosphatase, protease inhibitors and benzonase was used for cells lysis. One milligram of lysate was incubated overnight at 4 °C with 20 µl of Anti-Flag M2 magnetic beads (Sigma). After extensive washing in RIPA buffer, proteins were released in 2 × electrophoresis buffer and subjected to SDS–PAGE and western blotting.

Western blotting was performed using standard methods. Blots were incubated with primary antibodies: rabbit anti-WRN (Abcam); rabbit anti-DNA2 (Abgent); mouse anti-Tubulin (Sigma-Aldrich); rabbit anti-Lamin B1 (Abcam); mouse anti-DDK-Flag tag (Origene); rabbit anti-phosphoS4/8RPA32 (Bethyl); mouse anti-RPA32 (Calbiochem); mouse anti-CtIP (Santa Cruz); mouse anti-MRE11 (Abcam); rabbit anti-RAD51 (Santa Cruz); rabbi anti-phosphoS1133WRN (Genscript—custom); rabbit anti-phosphoS327CtIP (Thermo Fisher Scientific); rabbit anti-Histone H3 (Novus); rabbit anti-GST (Calbiochem); and goat anti-EXO1 (Santa Cruz). Blots were detected using the western blotting detection kit WesternBright ECL (Advansta) according to the manufacturer's instructions. Quantification was performed on scanned images of blots using Image Lab software, and values shown on the graphs represent normalization of the protein content evaluated through Lamin B1 or Tubulin immunoblotting. Detailed information on antibodies and their usage can be found in Supplementary Information Online. Uncropped images of the blots are shown in Supplementary Fig. 17.

**Immunofluorescence assays.** Cells were cultured onto 22 × 22 coverslip or eight-well Nunc chamber slides. To detect MRE11, WRN, BLM, RPA32 or pS4/8-RPA32

foci, we performed pre-extraction for 5 min on ice in CSK buffer followed by fixation with 4% PFA/PBS for 10 min, permeabilization in 0.5% Triton X100/PBS for 15 min and blocking in 10% FBS/PBS for 1 h, as described elsewhere[22]. Cells were incubated with specific primary antibody: rabbit anti-WRN (Abcam); rabbit anti-phosphoS4/8RPA32 (Bethyl); mouse anti-RPA32 (Calbiochem); mouse anti-MRE11 (Abcam); and goat anti-BLM (Santa Cruz), for 2 h at room temperature diluted in 1%BSA/PBS, followed by species-specific fluorescein-conjugated secondary antibodies (Alexa Fluor 594 Anti-Rabbit or Alexa Fluor 488 Anti-Mouse), and counterstained with $0.5 \mu g\,ml^{-1}$ 4,6-diamidino-2-phenylindole (DAPI). Slides were analysed with Eclipse 80i Nikon Fluorescence Microscope, equipped with a VideoConfocal (ViCo) system. For each time point, at least 100 nuclei were scored at $\times 40$. Detailed information on antibodies and their usage can be found in Supplementary Information Online.

**DNA fibre analysis**. DNA fibres were prepared, spread out and immunodecorated as previously described[20]. Briefly, cells were pulse-labelled with $25 \mu M$ 5-chloro-2′-deoxyuridine (CldU) and $250 \mu M$ 5-iodo-2′-deoxyuridine (IdU) at specified times, with or without treatment as reported in the experimental schemes. To prepare DNA fibers, $2.5 \mu l$ of cell resuspension ($10^6$ cells ml$^{-1}$ in PBS) were spotted onto cleaned glass slides and lysed with $7.5 \mu l$ of lysis buffer (0.5% SDS in 200 mM Tris-HCl, pH 7.4, 50 mM EDTA). After 6 min, slides were tilted (about 15°), allowing lysed cells to run down the slide slowly. Slides were air dried, fixed in methanol/acetic acid (3:1) and stored until immunofluorescence. For immunodetection of labelled tracks the following primary antibodies were used: anti-CldU (rat-monoclonal anti-BrdU/CldU; BU1/75 ICR1 Abcam; 1:100) and anti-IdU (mouse-monoclonal anti-BrdU/IdU; clone b44 Becton Dickinson, 1:10). The secondary antibodies were: goat anti-mouse Alexa Fluor 488 or goat anti-rabbit Alexa Fluor 594 (Molecular Probes, 1:200). The incubation with antibodies was performed in a humidified chamber for 1 h at room temperature. Images were acquired randomly from fields with untangled fibres using Eclipse 80i Nikon Fluorescence Microscope, equipped with a ViCo system. The length of labelled tracks were measured using the Image-Pro-Plus 6.0 software. A minimum of 100 individual fibres were analysed for each experiment and the mean of at least three independent experiments presented.

**Detection of nascent ssDNA**. To detect nascent ssDNA, cells were plated onto $22 \times 22$ coverslips in 35 mm dishes. After 24 h, the cells were labelled for 15 min before the treatment with $250 \mu M$ IdU (Sigma-Aldrich), cells were then treated with CPT $5 \mu M$ for different time points. Next, cells were washed with PBS, permeabilized with 0.5% Triton X-100 for 10 min at 4 °C and fixed with 2% sucrose and 3% PFA. For ssDNA detection, cells were incubated with primary mouse anti-IdU antibody (Becton Dickinson) for 1 h at 37 °C in 1%BSA/PBS, followed by Alexa Fluor488-conjugated goat-anti-Mouse and counterstained with $0.5 \mu g\,ml^{-1}$ DAPI. Slides were analysed with Eclipse 80i Nikon Fluorescence Microscope, equipped with a ViCo system. For each time point, at least 100 nuclei were scored at $\times 60$. Parallel samples either incubated with the appropriate normal serum or only with the secondary antibody confirmed that the observed fluorescence pattern was not attributable to artefacts. Fluorescence intensity for each sample was then analysed using ImageJ software. Detailed information on antibodies and their usage can be found in Supplementary Information Online.

**In situ PLA**. The in situ PLA in combination with immunofluorescence microscopy was performed using the Duolink II Detection Kit with anti-Mouse PLUS and anti-Rabbit MINUS PLA Probes, according to the manufacturer's instructions (Sigma-Aldrich).

To detect proteins, we used rabbit anti-WRN (Abcam) and mouse anti-MRE11 (Abcam) antibodies. Detailed information on antibodies and their usage can be found in Supplementary Information Online.

**HREJ reporter assay**. HEK293TshWRN cells were seeded in six-well plates at a density of 0.5 million cells per well. The next day, pCMV-FlagRnai-resWRN$^{WT}$, pCMV-FlagRnai-resWRN$^{S1133A}$ or pCMV-FlagRnai-resWRN$^{S1133D}$ were cotransfected with the I-SceI expression vector pCBASceI and the pHPRT-DRGFP plasmid reporter using DreamFect (OZ Biosciences) according to the manufacturer's instruction. Alternatively, the I-SceI linearized NHEJ reporter[40] replaced the pHPRT-DRGFP plasmid. pHPRT-DRGFP and pCBASceI were a gift from Maria Jasin (Addgene plasmids #26476 and #26477). Protein expression levels were analysed by western blotting 72 h post transfection. Cells were subjected to flow cytometry analysis at 72 h after transfection to determine the percentage of GFP-positive cells from $1 \times 10^5$ events. After correction for background level of GFP-positive cells obtained from no-I-Sce-I transfected cells, results were reported as percentage of repair efficiency compared with the number of GFP-positive cells observed in the presence of wild-type WRN.

**Chromosomal aberrations**. WRN$^{WT}$, WRN$^{S113A}$ and WRN$^{S1133D}$ cells were treated with $1 \mu M$ CPT (Sigma-Aldrich) at 37 °C for 4 h and allowed to recover for an additional 24 h. Cell cultures were incubated with colcemid ($0.01 \mu g\,ml^{-1}$) at 37 °C for 3 h until collecting. Cells for metaphase preparations were collected and prepared as previously reported[26]. For each condition used for treatment, chromosomal aberrations were examined in Giemsa-stained metaphases under a microscope (Leica) equipped with a charge-coupled device camera (Photometrics). For each time point, at least 100 chromosomes were examined by two independent investigators and chromosomal damage scored at $\times 100$.

**Statistical analysis**. All the data are presented as means of at least two independent experiments. Statistical comparisons of WS or WRN-mutant cells to their relevant control were analysed by analysis of variance or Mann–Whitney test. $P < 0.05$ was considered as significant.

**Data availability**. The data that support the findings of this study are available from the corresponding author on request.

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

## Acknowledgements

We are grateful to Dr Alessandra De Masi (Università di Roma Tre, Roma, Italy) for providing the NHEJ reporter plasmid, and to Dr Alessandro Vindigni (University of St. Louis, USA) for providing the siDNA2 oligos. We thank Dr Filomena Mazzei for critical reading of the manuscript. This work was supported by Fondazione Telethon to P.P. (grant no. GGP12144) and, in part, by investigator grants from Associazione Italiana per la Ricerca sul Cancro (AIRC) to P.P. (IG17383) and to A.F. (IG11871), and by the National Institutes of Health, NIA, Intramural Research Program to R.M.B. Jr.

## Author contributions

V.P. performed the analysis of WRN phosphorylation *in vivo* and the majority of the functional analysis of the CDK1-phosphorylation WRN mutants; P.P. performed immunoprecipitation for MS/MS analysis and radioactive kinase assay; S.R. performed MS/MS analysis and phosphorylation sites identification; M.S. performed the quantification of HR and NHEJ efficiency by flow cytometry; F.G. performed viability assays, neutral comet assay and biochemical analysis of chromatin accumulation of RPA32; J.A.S. performed WRN purification and enzymatic assays; L.Z. supervised the MS/MS analysis; V.P., M.S. and S.R. analysed data and contributed to designing the experiments; J.A.S, R.M.B. Jr, P.P and A.F. designed experiments and analysed data; P.P. and A.F. wrote the paper.

## Additional information

**Competing financial interests:** The authors declare no competing financial interests.

