## [Peer review file · Nature Communications]

Reviewers' comments:

Reviewer #1 (Remarks to the Author):

This work addresses an important problem. At DSBs, the regulation of end processing is very important for maintaining genome stability. While the regulation of short range resection has been well studied, long range resection is not as well understood. WRN protein, defective in Werner syndrome, participates in long range resection along with DNA2, and therefore WRN regulation may contribute to the overall control of resection in the cell. To study this, this work investigates the potential role of CDK phosphorylation of WRN. Mass spec study reveals that WRN is phosphorylated on S1133, within a CDK consensus motif. Non-phosphorylatable WRN S1133A is defective in resection at camptothecin induced, replication coupled DNA damage and the phosphomimic S1133D over-resects as shown by a fork protection assay. Importantly, S1133A alters repair pathway choice, reducing HR and increasing NHEJ, and S1133A shows genome instability in the form of gaps and radials in metaphase chromosomes. These are interesting studies, but there is one major experimental weakness:

Figure 1:

The major weakness is that the evidence that the effect of the S1133 mutations is due to CDK phosphorylation is incomplete. WRN undergoes multiple phosphorylations and many of these seem to be recognized by the antibodies used in Figure 1, which recognize S/TP sites, generic CDK sites, rather than a specific phosphopeptide. Because the antibodies are not specific, CDK inhibitors do not knockout phosphorylation, although they may reduce it, and neither does mutation of S1133 to non-phosphorylatable A, when studied by IP. The effect of inhibitors of CDK was not studied with mass spec. A more specific antibody is needed to insure that S1133 is a CDK site, i.e. an antibody to the phosphopeptide including pS1133. An siRNA knockdown of CDK1 and 2 should be carried out if CDK inhibitors do not remove response to the antibody. A demonstration of cell cycle specificity could be performed with more specific antibody which would increase confidence that phosphorylation is regulatory. According to their model, phosphorylation might also change upon treatment of cells with camptothecin, but this experiment is not performed, and needs to be carried out. Because of the ambiguity in the IP/westerns, all of the phenotypes that accompany S1133A or S1133D mutations could therefore be due to alterations in protein structure rather than phosphorylation or phosphorylation mediated changes. This caveat is real because the levels of the mutant proteins are lower compared to wildtype in many of the experiments. Another good experiment would be to show whether phosphorylation or S1133A or S1133D affect WRN helicase or nuclease activity. Demonstration that the enzymatic activities of WRN are intact in the S1133A phosphorylation site mutants is needed at the least. The in vitro phosphorylation by CDK is clear but the in vivo experiments are ambiguous.

The remainder of the extensive analyses use state of the art experiments and are well done, though it is sometimes difficult to understand the text provided to interpret them.

6.2.2. WS should be defined. Is this line helicase or nuclease deficient or both?

Suupl. Figure 2D: The reduction in P-RPA is not very convincing given the variation in RPA levels. The difference between reduction in HR and stimulation of NHEJ is an important result. Why is the NHEJ shown only in Supplemental Data?

It is interesting that WRN S1133A is still recruited to DNA damage foci. A better discussion of the reasons that Mre11 and WRN colocalize but that Mre11 and the WRN S1133A mutant do not colocalize is needed. The text is confusing. It seems like Mre11 should be required for WRN recruitment and not vice versa? Is there an explanation?

Reviewer #2 (Remarks to the Author):

In this study the authors identified a novel site of phosphorylation on WRN protein by CDK1 kinase at residue S1133. They conducted a series of experiments to uncover the role of CDK1 phosphorylation of WRN protein in recovery from replication induced DNA double strand breaks. By

expressing WRN phosphorylation mutants at this site (S1133A and S1133D) in human cells they provide evidence that CDK1 phosphorylation of WRN has a role in mediating long range end resection by WRN and DNA2 to promote repair by homologous recombination as opposed to NHEJ. Furthermore, phosphorylation of WRN at this site has a role in complex formation with the MRE11 complex. The manuscript is generally well written, although there are a few grammatical and typographical errors that require correction. The experiments support the conclusions and the model presented at the end of the manuscript reasonably well. However, there are several issues that require addressing in order to strengthen the study.

1. Figure 1A. It is important to show the corresponding W blot to prove the 180 kDa protein is WRN as outlined in the box.
2. Figure 1B confirms that the C-terminal fragment of WRN can be phosphorylated by CDK2-CyclinA, but does not show that residue S1133 is phosphorylated in this experiment. Please clarify. Is S1133 the only possible CDK phosphorylation site in the WRN C-terminal fragment?
3. Pg. 5 bottom. It looks like the WRN S1133A mutant is not expressed as well as WT WRN based on the W blot (bottom panel). Is this mutant protein less stable?
4. Fig. 1. Statistical analyses should be done for the bar graphs in Fig. 1 D and E.
5. Pg. 6 top. The authors conclude that the residual anti-CDK substrate reactivity observed in the presence of inhibitor suggests that the antibody may not be very specific. An alternative explanation is that the CDK inhibitors are not 100% effective. Other substrates of CDK1 could be tested as a positive control.
6. Fig. 2. The authors should explain how IdU incorporation is specific for the leading nascent strand. IdU should also be incorporated on the lagging nascent strand. Statistical analysis should be included for the bar graph in panel C.
7. Fig. 3. The authors should comment on what is known regarding the specificity of the WRN helicase inhibitor used. Are other RecQ helicases such as BLM affected? Is this known? Has this been rigorously tested? Could an siWRN be used as shown later in the study?
8. Supp Fig 7. The accumulation of 53BP1 foci does not necessarily indicate that NHEJ is occurring. Two previous reports showed that 53BP1 foci are induced by replication stress and likely represent regions of incomplete DNA synthesis. They could be sites of failures in restarting DNA replication (Harrigan et al, JCB, 2011, v 193, p. 97; Lukas et al, Nat Cell Biol, 2011, vol 13 p 243).
9. Fig 5. Why do some of the cells in panel A lack WRN staining? Cells that do not show WRN in foci should at least show WRN expression, and should show WRN localization in the nucleoli based on previous studies.
10. Fig 5. Panels D and E, the image for the critical control of unperturbed conditions should be included. No PLA positive foci should be detected here since WRN and MRE11 are not expected to interact. The authors mention they did this control, but they have not shown the data.
11. Figure 7. The survival data is confusing. It appears that WS cells are less sensitive to CPT than wild type cells at 0.1 μ M and 5 μ M CPT, and slightly more sensitive at 1 μ M CPT. However, there is a wealth of literature showing that WS cells are hypersensitive to CPT. This makes it difficult to interpret the data with the phosphorylated WRN mutants.
12. Fig. 6. If the WRNS1133D cells are truly hyper-recombinogenic, then one would expect to see an elevation of SCEs. This would provide stronger evidence for a hyper-recombinogenic phenotype than a plasmid based report assay.

13. There is some redundancy in the discussion. For example, the report that DNA2 works with WRN and BLM to mediate long range resection is mentioned twice in the second paragraph of the discussion.

14. The reliance on pharmacological inhibition of RAD51 and DNA-PKcs for the experiments in Figure 7 is somewhat concerning, in the absence of any genetic manipulation of RAD51 or DNA-PKcs. For example, the expression of the dominant negative form of RAD51 would strengthen the data. At the very least, the authors should speak to the specificity and efficacy of these inhibitors.

Reviewer #3 (Remarks to the Author):

A. This work proposes a novel function for WRN helicase S1133 residue phosphorylation in regulating long range end resection at DSBs induced by camptothecin treatment of upon generation of a single DSBs in mammalian cells. S1133 phosphorylation is also proposed to regulate the choice between HR and NHEJ. The authors claim that WRN S1133 phosphorylation by CDK1 is a trigger for WRN interaction with Mre11 and localization of this latter at DSBs. The main conclusions of this work stem from the use of a mammalian cell line defective in the WRN helicase (WS) complemented by stable expression wild-type or WRN variants bearing point mutations in the WRN S1133 residue, followed by analysis of the corresponding phenotypes.

B. The function of WRN S1133 phosphorylation in DNA repair has not been previously explored, hence it is novel.

C. Overall the work is well performed, and several results presented in the manuscript are convincing. The methodology employed is appropriate and in line with methods currently used to analyze resection, repair and replication fork movement. The quality of the data is of good standard. Notwithstanding, the data must be strengthened in several parts of the manuscript by including appropriate controls (see section F).

D. The data replicate are robust and the statistics applied to the experimental sets are appropriate.

E. The conclusion that phosphorylation of WRN S1133 residue regulates long-range resection and DNA repair pathway choice is convincing and robust. The phenotype observed upon complementation of WS cells with the corresponding WRN S1133 mutants are reliable and reproducible in the author's hands. However, the main conclusion that CDK1 is the kinase that phosphorylates WRN S1133 residue is not at all convincing. Hence, if the authors wish to sustain this claim they must strengthen the experimental data to convincingly show that this is the case (see below).

F. In general, in the figure legends of the panels showing immunofluorescence data, it is not mentioned that (I imagine) the blue colour corresponds to DAPI or Hoechst (?) staining. This should be clearly stated.

In the cartoon of Fig. 2A, why is drawn labelling one replicated strand? The other strand should be labelled too. In the figure legends of Immunofluorescence panels it is not indicated at which time points the images were taken.

1. The MS/MS data relatives to the enrichment of S1133 phosphopeptide after genotoxic treatments are not presented as a whole in Fig. 1, which weakens the author's claim. Also, how do the authors know that the excised band correspond to WRN before excision and MS/MS analysis? Consensus sites for CDK1 phosphorylation are far to be well defined. Indeed, although authors indicate that the S1133 is a perfect match to CDK1 they show in Fig. 1 that this site is phosphorylated by CDK2 and somehow sensitive to roscovitine, which is a more specific CDK2

inhibitor. In addition, as also suggested by the authors, this site could also be an ATM substrate (or a DNA-Pk). Hence, the claim that S1133 is a specific CDK1 phosphorylation site is very weak and is not at all demonstrated, as the authors claim in the manuscript. Since this issue is the main claim of the manuscript, authors must investigate it in more detail by performing additional experiments, including, but not limited to, CDK1, CDK2 siRNA, ATM and/or ATR inhibitors. Further, a low exposure of the autoradiogram shown in Fig. 1C should be shown to appreciate WRN phosphorylation. The authors should also explain the apparent CDK2 autophosphorylation visible in this panel. In the Coomassie stain of the same panel, in the lane containing WRN and CDK2/GST-CycA there are polypeptides that are not present in the lanes loaded with GST-CycA alone: what are these? A lane loaded with WRN alone is required. The result shown in Fig. 1D is not convincing. The claimed decrease in WRN phosphorylation is barely detectable. An immunoprecipitation with non-specific IgG should also be included to show the specificity of the anti-Flag immunoprecipitation. The experiment shown in Fig. 1E is unacceptable under the present form. The samples should be loaded in the same gel to be compared. If the figure comes from the same cropped gel, then the entire gel should be shown. In particular, this panel shows that the CDK1 inhibitor reduces but does not eliminate WRN phosphorylation, suggesting that kinases other than CDK1 may also phosphorylate WRN on S1133 residue.

2. The inclusion of untreated (-CPT) cells in Fig. 2B and 3C would strengthen the data. The effect of WRN mutants on the recruitment of RPA (not phosphorylated) at DSBs by immunofluorescence and western blot should be also addressed. This is an important point since RPA is strongly implicated in long end resection in both yeast (Chen et al., Mol Cell 2013) and *Xenopus* (Tammaro et al., Nucl. Acids Res. 2015), and it is part of the mechanism explored by the authors. These works are not cited in the manuscript.

3. Analysis of resection in WS cells with or without WRN inhibitor should be included to show the specificity of the inhibitor. The western blot shown in Fig. 3D is unacceptable under the present form. A whole cell extract with a loading control and WRN blot should be shown to appreciate the efficiency of siRNA EXO1. Then a second panel showing the immunoprecipitation should be included showing the IgGs and WRN levels.

4. The length of the DNA fibers analyzed in all the experimental conditions must be shown in Figure 4.

5. A blot with WRN should be shown in Fig. 5B.

6. In Fig. 6C, the claim that Rad51 accumulates onto chromatin upon CPT treatment is unconvincing. First, cytoplasmic and chromatin samples obtained during the fractionation procedure showing an enrichment of Rad51 onto chromatin and of that of chromatin-bound proteins (histones or the Origin Recognition Complex, ORC) should be shown in the same gel. Then, the claimed enrichment of Rad51 must be shown using low exposures of the western blot and quantified against at least two different loading controls.

Fig. 6D, the sentence « in wild-type cells, the level of DSBs induced by CPT dropped of more than 50% after recovery » does not sustain the observed data shown in Fig. 6D, where a level lower than 50% is observed. However this appears to be the case in the phosphomimetic mutant. Also, this latter does not seem to stimulate significantly repair after recovery when compared to cells complemented with wild-type WRN. This must be corrected and the interpretation of these data must be revised.

7. Materials and Methods. The reference of the antibodies used in the manuscript must be indicated. The experimental procedure to produce recombinant WRN and CDK2/Cyc-A complex is not described.

G. Previous work is correctly cited except for the following two papers that have not been cited showing an implication of RPA with WRN both in yeast and *Xenopus* respectively: Chen et al., Mol

Cell 2013, Tamaro et al., Nucl. Acids Res. 2015, .

H. The paper is clearly and lucidly written, making the wright point on the main issue of the work. The writing can be improved in several places by correcting several spelling mistakes upper versus lower case for gene names, using appropriate expressions for describing experimental procedures and data, and explaining the experimental set up. The way the paper is written is too specialized (see some suggestions below). I suggest the paper to be proofread by an english native speaker.

The authors should explain that WS cell line is mutated in WRN, which is fundamental to interpret the results.

Page 5, line 9, the sentence "...treated cultures with nothing.. » should be changed to « untreated cultures » or to a similar sentence.

Page 7, fourth line of the second paragraph, « after 30 minutes of CPT, consistently... » change to « after 30 minutes of CPT treatment, consistent... ». Last paragraph, first line, change « the WRN helicase activity supports the DNA2 exonuclease... » to « WRN helicase stimulates DNA2 exonuclease activity ... ». Line 6 change « consistently ... » to « consistent... ».

The title and the first line of the last paragraph of page 8 are unclear.

Page 9, first line, the significance of the experimental set up must be clearly described (why replication forks are pulse- abeled with IdU/CdU ?). Last line, please explaine what PLA is. The same must be also done for the last paragraph of page 10. Please explain the meaning of detection of 53BP1 foci.

Page 11, explain the rational of the COMET assay.

Discussion, "change unphosphorylable" to "unphosphorylatable".

The effect of WRN S1133A mutant on the recruitment of BLM suggested in the discussion can be shown experimentally.

Point-by-point response to reviewers

Reviewer #1

We would like to thank the reviewer for the appreciation of our work and for the insightful comments. In these last few months, we worked hard to provide additional data supporting our claim of a CDK-dependent phosphorylation of WRN S1133. We could take advantage from a specific, custom-made, antibody generated against a phosphopeptide of WRN that contains pS1133. We repeated all the experiments aimed to support CDK-dependent phosphorylation of WRN S1133 using this specific pS1133WRN antibody and results are now provided in the revised ms (Figure 2, Supplementary Figure 2 and 3). We hope that these new results contributed to strengthen our conclusions. Moreover, we also carefully revised the text, also with the assistance of one of the co-authors, who is a native English speaker, to make interpretation of the results easier.

Below our point-by-point answer:

The major weakness is that the evidence that the effect of the S1133 mutations is due to CDK phosphorylation is incomplete. WRN undergoes multiple phosphorylations and many of these seem to be recognized by the antibodies used in Figure 1, which recognize S/TP sites, generic CDK sites, rather than a specific phosphopeptide. Because the antibodies are not specific, CDK inhibitors do not knockout phosphorylation, although they may reduce it, and neither does mutation of S1133 to non-phosphorylatable A, when studied by IP.

Actually, the CDK motif antibody originally used in our work was a little more specific than the standard anti-S/TP antibody generally used in other works. According to the vendor, the motif antibody is claimed to recognize the PXPSP or pSPXR/K sequence, which is only present once in WRN, around S1133. However, as stated in the original version of our ms, this antibody probably recognized other phosphorylation sites or, most likely, retained an ability to bind to the unmodified WRN sequence that is not negligible under our experimental conditions.

To deal with this drawback, we obtained a custom-made antibody specifically directed against a WRN phospho-peptide containing pS1133. With this reagent in hands, we repeated all the experiments investigating CDK-dependent phosphorylation of WRN *in vivo* and *in vitro*, which are now reported in Figure 1 and in Supplementary Figures 2 and 3 of the revised version of our ms. Results convincingly demonstrate that:

1. S1133 of WRN is phosphorylated in a CDK-dependent manner (Figure 1C; Supplementary Figure 3A and B). Indeed, treatment with the pan-CDKs inhibitor roscovitine results in 90-97% of reduction of the anti-pS1133WRN WB signal; the same reduction observed after dephosphorylation of the IP with λ -PPase (Figure 1C, lane 3 vs. lane 4);
2. S1133 of WRN is phosphorylated in a CDK1-dependent manner (Figure 1D; Supplementary Figure 3A). As the reviewer will see, treatment with the CDK1-specific inhibitor RO-3306 results in an almost complete (From 95 to 100%) reduction of the anti-pS1133WRN WB signal; for comparison, reduction observed after dephosphorylation of the IP with λ -PPase is 97% (Figure 1D, lane 3 vs. 5 vs. 6);
3. Detection of S1133 phosphorylation by anti-pS1133WRN is almost-completely abrogated (94%) in the WRN-S1133A mutant (Figure 1D), and the residual signal is comparable to what observed after dephosphorylation of the WRN-WT IP (Figure 1D, lane 5 vs. 6);
4. Phosphorylation of WRN at S1133 is strongly stimulated by CPT treatment. Indeed, using the specific anti-pS1133WRN antibody, we observed a 3-fold and more increase in the level of WRN phosphorylation at S1133 (Supplementary Fig. 2B);
5. Phosphorylation of S1133 is detected by anti-pS1133WRN WB after *in vitro* phosphorylation of a C-terminal fragment of WRN with a purified CDK2/CycA complex (Suppl Fig. 2B).

For what concerns the very low level of residual signal observed in anti-pS1133WRN blots, when detected, it has comparable intensities to what observed in the internal λ -PPase control. Thus, it seems likely that such variable and nevertheless very low level of residual signal on anti-pS1133WRN WBs can be ascribed to recognition of the S1133-unphosphorylated WRN sequence. This is not uncommon, and can be related to the fact that only a small fraction of WRN is phosphorylated at S1133 at any given time. Alternatively, CDKs inhibitors could be not 100% effective in inhibiting the kinases. However, at least for what concerns roscovitine, the concentration used in our experiments appears to be effective as almost-completely abrogated phosphorylation of CtIP at S327, a CDK site (Suppl. Fig. 4).

We are confident on our results and we think that such a residual signal does not affect our conclusions.

The effect of inhibitors of CDK was not studied with mass spec. A more specific antibody is needed to insure that S1133 is a CDK site, i.e. an antibody to the phosphopeptide including pS1133.

We also performed MS/MS experiments using anti-WRN IP from roscovitine-treated cells, however, as phosphopeptide identification is performed after phosphopeptide enrichment by TiO₂ columns, the instrument failed to detect any peptide containing S1133. Thus, we were not able to include any spectra from roscovitine-treated samples. It should be kept in mind that without phosphopeptide enrichment it was not possible to identify WRN peptide containing pS1133. As better explained earlier on, we now presented experiments using a custom-made rabbit anti-pS1133WRN antibody that should help in confirming actual CDK-dependent phosphorylation.

An siRNA knockdown of CDK1 and 2 should be carried out if CDK inhibitors do not remove response to the antibody. A demonstration of cell cycle specificity could be performed with more specific antibody which would increase confidence that phosphorylation is regulatory.

In Figure 1 and Supplementary Figures 2 and 3, we now show that treatment with phosphatase of the IP, treatment with CDK inhibitors or expression of the unphosphorylatable S1133A-WRN mutant similarly suppresses detection by pS1133WRN antibody. Moreover, we provide a proof that the concentration of roscovitine used in our experiments is sufficient to abrogate detection of phosphorylation of CtIP at S327, a CDK target, by a commercial anti-pS327CtIP antibody. As we had evidence that small-molecule inhibition of CDKs was efficient, we preferred to avoid depletion of CDK2 and 1 by RNAi as, at least in our hands, it caused S-phase alterations and would have affected the analysis of an S-phase-specific event such as processing of replication-dependent DSBs. The same applies for the cell cycle specificity asked for the reviewer. Indeed, the focus of our work was the role of CDK-dependent phosphorylation in the context of replication-dependent DSBs. As such, we judged that it was much more relevant presenting evidence supporting the correlation between these two events, rather than analyzing S1133

phosphorylation during cell cycle. We think that this interesting point could be better investigated in a follow-up paper. It should be also noted that the ms already contains eight multi-panel figures and sixteen supplementary figures in its current form, so it would be hard to efficiently introduce and discuss additional results.

According to their model, phosphorylation might also change upon treatment of cells with camptothecin, but this experiment is not performed, and needs to be carried out.

This is a crucial point indeed. We now presented IP/WB data demonstrating that treatment with CPT increases the level of S1133 phosphorylation of more than 3-fold. Interestingly, it should be noted that greater is the level of S1133 phosphorylation easier is the discrimination between phosphorylated and unphosphorylated fraction of WRN by the antibody (See Suppl. Fig. 3B).

Because of the ambiguity in the IP/westerns, all of the phenotypes that accompany S1133A or S1133D mutations could therefore be due to alterations in protein structure rather than phosphorylation or phosphorylation mediated changes. This caveat is real because the levels of the mutant proteins are lower compared to wildtype in many of the experiments.

Actually, the protein levels of the wild-type WRN and the two phosphorylation mutants are pretty similar, both in the stable cell lines and in transiently-transfected HEK293T. This information can be easily obtained from any of our WB. We analysed whether the two phosphorylation WRN mutants could undergo to enhanced protein degradation compared to the wild-type using the proteasome inhibitor MG132, and the result of our experiment is now provided as additional figure for the reviewers at the end of the text. Thus, we can exclude that mutations do not affect protein stability.

Another good experiment would be to show whether phosphorylation or S1133A or S1133D affect WRN helicase or nuclease activity. Demonstration that the enzymatic activities of WRN are intact in the S1133A phosphorylation site mutants is needed at the least.

This is an interesting point. Although our new experiments provide us stronger proofs supporting the phosphorylation of WRN S1133 by CDK, this is a nice suggestion. We asked for collaboration to Dr. Brosh, who is one of the leading scientist in the biochemistry of WRN and other RecQ helicases, and now provide additional data on the enzymatic activity of the phosphorylation WRN mutants (Figure 3B and C). They show that mutations do not affect at all the *in vitro* ability of WRN to unwind a forked model substrate or, most importantly, to degrade a model duplex substrate.

The in vitro phosphorylation by CDK is clear but the in vivo experiments are ambiguous.

As thoroughly reported in our answers above, the new experiments performed using the custom-made anti-pS1133WRN specific antibody strengthen our claim the WRN

is phosphorylated by CDK at S1133, and that this event is stimulated upon CPT treatment.

Minor comments

6.2.2. WS should be defined. Is this line helicase or nuclease deficient or both?

We now included, in materials and methods, a sentence about the *WRN* mutation in the WS cell line. The cells we used actually are natural KO, as the mutation introduces a premature stop codon well before the NLS, and the truncated protein is retained in the cytoplasm and degraded; as such no *WRN* protein is present in our cell line (as it occurs essentially in all the WS-derived cells).

Suppl. Figure 2D: The reduction in P-RPA is not very convincing given the variation in RPA levels.

The blots presented in that supplementary figure does not come from a WCE. It is from a chromatin fraction. Thus, variation in the level of RPA32 relates with its accumulation in chromatin, and is another readout of ssDNA formation, either following fork stalling or because of end-resection. Levels of p-RPA should be normalized using both RPA32 and LMNB1. Now, as also requested by another reviewer, we included the analysis of the accumulation of RPA32 foci (total protein, not only the S4/8 phosphorylated one). Basically, all our assays pinpointing, directly or indirectly, formation of ssDNA consistently show that abrogation of S1133 phosphorylation affects end-resection, and that S1133A mutation and concomitant CDK inhibition do not induce any additive effect.

The difference between reduction in HR and stimulation of NHEJ is an important result. Why is the NHEJ shown only in Supplemental Data?

We now included the NHEJ assay in the main figures. As a consequence, also for sake of limit in the figure size, we moved the analysis of RAD51 chromatin accumulation in the supplementary figures

It is interesting that WRN S1133A is still recruited to DNA damage foci. A better discussion of the reasons that Mre11 and WRN colocalize but that Mre11 and the WRN S1133A mutant do not colocalize is needed. The text is confusing. It seems like Mre11 should be required for WRN recruitment and not vice versa? Is there an explanation?

We now included a short sentence in the attempt to clarify this point. However, a possible explanation for this findings was already present in the original discussion. Our explanation is that *WRN* and *MRE11* co-localise and interact because they work together during end-resection. As the *WRN*-S1133A mutant apparently interferes with long-range end-resection causing a pathway switch, loss of interaction is a

consequence of dismantling of MRE11 from collapsed forks to make room for End-Joining proteins. Indeed, MRE11 foci are also reduced. So, in a wild-type context, MRE11 is required for WRN recruitment, but when WRN cannot be phosphorylated other interactions, or even PTMs, may be required to retain WRN at damaged sites.

Reviewer #2

We appreciated that our work has been positively evaluated by the reviewer. We tried to improve our work according to comments and suggestions raised by the reviewer. In particular, we included a new set of experiments to demonstrate that S1133 of WRN is phosphorylated *in vivo* and *in vitro* by CDK using an antibody specifically-generated to detect WRN phosphorylated at S1133, added analysis of SCEs in the phosphorylation mutants of WRN, and the analysis of viability in S1133D-WRN-expressing cells transiently-transfected with a dominant-negative form of RAD51 (SM-RAD51).

Below, the reviewer will find a point-by-point answer:

1. Figure 1A. It is important to show the corresponding W blot to prove the 180 kDa protein is WRN as outlined in the box.

Unfortunately, we did not save any material for WB analysis when we prepared samples for MS/MS. However, samples used for Mass spec were from immunoprecipitation of transiently-expressed Flag-tagged WRN using anti-Flag antibody. As such, it is unlikely that the prominent 180KDa protein visible in the CBB-stained gel only in the transfected samples and not present in the mock-transfected sample (lane 1), may be another protein. Moreover, MS/MS analysis did not find peptides attributable to other proteins, and the phosphopeptide identified can be only attributable to WRN. That said, we repeated MS/MS analysis, as requested also by another reviewer, to show that phosphopeptides containing S1133 can be identified also after CPT treatment, and we performed anti-WRN WB on a small fraction of the immunoprecipitated material. The WB, presented in Supplementary Figure 4, confirms that the 180KDa protein actually is WRN.

2. Figure 1B confirms that the C-terminal fragment of WRN can be phosphorylated by CDK2-CyclinA, but does not show that residue S1133 is phosphorylated in this experiment. Please clarify. Is S1133 the only possible CDK phosphorylation site in the WRN C-terminal fragment?

Actually, S1133 is the only putative CDK site in the C-terminal fragment of WRN used as substrate in the *in vitro* kinase assay, as also stated in the introductory paragraph of the results of figure 1. To unambiguously confirm that CDK2/CyclinA

phosphorylate S1133, we now included the analysis by WB of the kinase assay using the custom anti-pS1133WRN antibody (Supplementary Fig. 2B).

3. Pg. 5 bottom. It looks like the WRN S1133A mutant is not expressed as well as WT WRN based on the W blot (bottom panel). Is this mutant protein less stable?

The level of expression of the phosphorylation mutants of WRN is not overtly different between them and the wild-type WRN, in the stable cell lines (Fig. 2A). The variation noticed by the reviewer is something related with transient transfection. We optimized transfection and, as the new experiments confirm, expression level of the S1133A mutant is similar to that of the wild-type. We included at the end of the point-by-point answer text, a WB from cells treated or not with MG132 to inhibit proteasome. This experiment confirms that the WRN mutants are not less stable than a wild-type protein.

4. Fig. 1. Statistical analyses should be done for the bar graphs in Fig. 1 D and E.

In the revised version of the ms, we omitted the graph and only include quantification of the representative WB obtained using the custom anti-pS1133WRN antibody. Now, differences are more straightforward and blots clearly show that S1133 of WRN is phosphorylated in a CDK-dependent manner (Figure 1, Supplementary Figure 2 and 3).

In some of the blots, a residual pS1133WRN-immunoreactive band is still present, however, when it occurs, the intensity is similar to that detectable after lambda-phosphatase treatment of the IP, or even less. Our explanation is that the fraction of S1133-phosphorylated WRN is not high, especially in untreated cells, and thus sometimes a signal corresponding to residual reactivity of the anti-pS1133WRN antibody to the unphosphorylated sequence can come up, in spite of our efforts to optimize antibody dilutions and amount of protein loaded into the gel. It should be noted, however, that after CPT treatment the fraction of S1133-phosphorylated WRN increases (Supplementary Figure 3B).

5. Pg. 6 top. The authors conclude that the residual anti-CDK substrate reactivity observed in the presence of inhibitor suggests that the antibody may not be very specific. An alternative explanation is that the CDK inhibitors are not 100% effective. Other substrates of CDK1 could be tested as a positive control.

In the revised ms, we repeated all the experiments using the custom anti-pS1133WRN antibody, that is an antibody generated against a peptide containing pS1133. As such, the results of the IP/WB are now much convincing, at least in our opinion. To test that the concentration of CDK inhibitor was effective, we analysed phosphorylation of a relevant CDK target, CtIP, at S327 by a commercially-available specific antibody (Supplementary Figure 4C). The WB shows that the dose of roscovitine used throughout our work is sufficient to abrogate almost completely detection of pS327 of CtIP. We also repeated this experiments using twice the

concentration of roscovitine, and we had comparable results, a strong reduction in the anti-pS327-CtIP signal, but for sake of clarity, we omitted that blot. We can provide to the reviewer the blot anytime, if needed.

6. *Fig. 2. The authors should explain how IdU incorporation is specific for the leading nascent strand. IdU should also be incorporated on the lagging nascent strand. Statical analysis should be included for the bar graph in panel C.*

Actually, IdU incorporation is not strand-specific. In the cartoon, only the leading nascent strand is shown for sake of clarity, but IdU is incorporated also in the lagging nascent strand. This was clarified in the revised results. Statistical analyses have been now included.

7. *Fig. 3. The authors should comment on what is known regarding the specificity of the WRN helicase inhibitor used. Are other RecQ helicases such as BLM affected? Is this know? Has this been rigorously tested? Could an siWRN be used as shown later in the study?*

The inhibitor is specific to the WRN helicase. The specificity and characterization of the inhibitor can be found described in the relevant, included, reference.

8. *Supp Fig 7. The accumulation of 53BP1 foci does not necessarily indicate that NHEJ is occurring. Two previous reports showed that 53BP1 foci are induced by replication stress and likely represent regions of incomplete DNA synthesis. They could be sites of failures in restarting DNA replication (Harrigan et al, JCB, 2011, v 193, p. 97; Lukas et al, Nat Cell Biol, 2011, vol 13 p 243).*

That is an interesting point. Formation of 53BP1 has been sometime used as an unspecific sign of NHEJ or EJ activation. However, more recently, as referred correctly by the reviewer, formation of 53BP1 nuclear bodies in G1 cells has functionally linked to persistence of unreplicated genomic regions, often at CFS. Actually, the 53BP1-positive cells observed in our experimental condition do not present nuclear bodies but rather a “focal” distribution. Moreover, since the IF analysis was carried out 90min post treatment, affected cells could not be in G1-phase. That said, as we agree that formation of 53BP1 is not a unique marker of EJ activation and, as we presented several other, and more specific, proofs of an HR-NHEJ switch in S1133A-expressing cells, we omitted the analysis from the revised ms.

9. *Fig 5. Why do some of the cells in panel A lack WRN staining? Cells that do not show WRN in foci should at least show WRN expression, and should show WRN localization in the nucleoli based on previous studies.*

As explained in the materials and methods section, we routinely perform WRN IF after an “in situ fractionation” step, as originally developed in the Petrini’s lab. This protocol involves extraction of nuclear-free protein, leaving only the fraction associated to chromatin. Thus, WRN is detectable only in foci as we get rid of the

unbound and nuclear-free fractions that usually make difficult to detect clearly foci formation. It should be noted that, at the best of our knowledge and experience, WRN is found in the nucleoli, however, using in situ fractionation, bright WRN-positive nucleoli are detected only in a subset of cell lines, such as HeLa, which have large nucleoli per se. Cells having small nucleoli, as ours, often show poor WRN staining. To confirm that mutations do not affect nucleolar WRN localization and to give proof of that, we included WRN IF images from untreated cells obtained at longer acquisition times in the revised ms (Supplementary Figure 11).

10. Fig 5. Panels D and E, the image for the critical control of unpertured conditions should be included. No PLA positive foci should be detected here since WRN and MRE11 are not expected to interact. The authors mention they did this control, but they have not shown the data.

This control is now included as Supplementary Figure 12.

11. Figure 7. The survival data is confusing. It appears that WS cells are less sensitive to CPT than wild type cells at 0.1 μ M and 5 μ M CPT, and slightly more sensitive at 1 μ M CPT. However, there is a wealth of literature showing that WS cells are hypersensitive to CPT. This makes it difficult to interpret the data with the phospho WRN mutants.

In the revised ms, sensitivity of cells to CPT was analysed by clonogenic survival. The new experiment shows that WS are sensitive to CPT, as described by several groups, including our.

12. Fig. 6. If the WRNS1133D cells are truly hyper-recombinogenic, then one would expect to see an elevation of SCEs. This would provide stronger evidence for a hyper-recombinogenic phenotype than a plasmid based report assay.

We analysed the formation of SCEs after CPT treatment. The results are now provided in Figure 7C. Expression of a phosphomimetic WRN-S1133D protein resulted in a 3-fold increase in SCEs.

13. There is some redundancy in the discussion. For example, the report that DNA2 works with WRN and BLM to mediate long range resection is mentioned twice in the second paragraph of the discussion.

We carefully checked the discussion and removed duplications; we also improved English.

14. The reliance on pharmacological inhibition of RAD51 and DNA-PKcs for the experiments in Figure 7 is somewhat concerning, in the absence of any genetic manipulation of RAD51 or DNA-PKcs. For example, the expression of the dominant negative form of RAD51 would strengthen the data. At the very least, the authors should speak to the specificity and efficacy of these inhibitors.

The DNA-PKcs inhibitor used is also widely and successfully used in many other groups and has been used in many publications, making us confident about its specificity. For what concerns the RAD51 inhibitor, we had the possibility to use it previously and, at the best of our experience, it recapitulates many of the effect observed with RAD51 RNAi without showing limitations of the long-term RAD51 depletion. That said, to strengthen our claim, we analysed the effect of the transient over-expression of a dominant-negative RAD51 chimaera (SM-RAD51) on the sensitivity to CPT originally. This construct was developed in the Lopez's group several years ago and it has been reported to interfere with nucleofilament formation and nucleation but not overtly affect viability. Results of this experiment, which confirm what observed using the RAD51i, are reported in Supplementary Figure 16A, B.

Reviewer #3

We would thank the reviewer for the appreciation of our work. We noticed that, while overall positive, the opinion of the reviewer about our work is weakened by the need to strengthen experimental data supporting that CDKs phosphorylate S1133 of WRN.

In these four months, we obtained a specific antibody directed against a WRN peptide containing pS1133. Using this anti-pS1133WRN antibody, we repeated all the IP/WB experiments and also the *in vitro* kinase assay. In addition, we also analysed if CPT treatment could stimulate S1133 phosphorylation. The outcome of our hard work, is provided in Figure 1, Supplementary Figure 2 and 3 of the revised ms.

As also illustrated in the point-by-point answer to the other two reviewers, these experiments confirm that S1133 phosphorylation is similarly sensitive to roscovitine and the CDK1 inhibitor, which suppressed detection by the anti-pS1133WRN antibody completely, or almost completely (Supplementary Figure 3). When a residual anti-pS1133WRN antibody reactivity is detected, it is not higher than that detectable on lambda phosphatase-treated anti-Flag(WRN)-immunoprecipitates, indicating that it is not due to other kinases involved in phosphorylating S1133 but rather in the variable level of cross-reactivity against the unphosphorylated WRN sequence.

F. In general, in the figure legends of the panels showing immunofluorescence data, it is not mentioned that (I imagine) the blue colour corresponds to DAPI or Hoechst (?) staining. This should be clearly stated.

We now indicated in the materials and methods that IFs were counterstained with DAPI.

In the cartoon of Fig. 2A, why IdU is drawn labelling one replicated strand ? The other strand should be labelled too. In the figure legends of Immunofluorescence panels it is not indicated at which time points the images were taken.

We depicted the leading nascent strand for sake of clarity. We now specified that point clearly. We still omitted to show lagging strand to do not affect readability of the drawing.

The time point is indicated below the IF panels (90').

1.The MS/MS data relatives to the enrichment of S1133 phosphopeptide after genotoxic treatments are not presented as a whole in Fig. 1, which weakens the author's claim.

Actually, phosphosite identification has been achieved after a series of enrichment steps using titanium oxide charged tipcolumns. The unphosphorylated peptides were lost in the flow-through. Thus, we cannot provide exact data about enrichment. Consistently, we modified our sentence in the results accordingly, omitting any quantitative indication. It should be noted by the reviewer that the IP/WB analysis of S1133 phosphorylation of WRN using the anti-pS1133WRN antibody confirmed that the fraction of phosphorylated WRN increases upon CPT treatment (Supplementary Fig. 3B). We hope that these new results contribute to reinforce our claim.

Also, how do the authors know that the excised band correspond to WRN before excision and MS/MS analysis?

At the best of our knowledge, it is unlikely that the specific 180kDa signal presents only in the anti-Flag IP from Flag-WRN-transfected cells may be other than WRN. Moreover, the peptide identified by MS/MS belongs to WRN and it was identified together with other phosphopeptides assigned by the software to WRN. However, to answer a similar question raised also by reviewer #2, we repeated the experiment, with the aim to confirm S1133 phosphorylation after CPT treatment by MS/MS, and in this experiment we used a small fraction of the immunoprecipitate to detect WRN by WB. The result is shown in Supplementary Figure 4B.

Consensus sites for CDK1 phosphorylation are far to be well defined. Indeed, although authors indicate that the S1133 is a perfect match to CDK1 they show in Fig. 1 that this site is phosphorylated by CDK2 and somehow sensitive to roscovitine, which is a more specific CDK2 inhibitor.

We stated that S1133 is a perfect match to the general CDK consensus sequence. As such, we reasoned that any CDKs/Cyclin complex would have targeted that site *in vitro*, and we used the CDK2/CycA complex simply because we could easily get it.

Our new data from IP/WB experiments and the use of the anti-pS1133WRN antibody, show that phosphorylation can be suppressed by the CDK1i as well as by the pan-specific (at the concentration used) CDK-inhibitor roscovitine.

In addition, as also suggested by the authors, this site could also be an ATM substrate (or a DNA-Pk). Hence, the claim that S1133 is a specific CDK1 phosphorylation site is very weak and is not at all demonstrated, as the authors claim in the manuscript. Since this issue is the main claim of the manuscript, authors must investigate it in more detail by performing additional experiments, including, but not limited to, CDK1, CDK2 siRNA, ATM and/or ATR inhibitors.

We did not find the sentence where it was stated that S1133 of WRN can be an ATM target. We simply stated that WRN can be targeted by other kinases, phosphorylation sites of which we contributed to define. However, as also shown in Ammazalorso et al., S1133 is not one of the ATM-targeted phosphorylation sites of WRN. However, the new results obtained using the anti-pS1133WRN specific antibody should contribute to support our claim about CDK-dependent phosphorylation, excluding involvement of other kinases, at least under our specific experimental setting.

Thus, we considered no more relevant to perform IP/WB experiments using RNAi to down-regulate CDK1, 2, ATM or ATR.

Further, a low exposure of the autoradiogram shown in Fig. 1C should be shown to appreciate WRN phosphorylation. The authors should also explain the apparent CDK2 autophosphorylation visible in this panel. In the Coomassie stain of the same panel, in the lane containing WRN and CDK2/GST-CycA there are polypeptides that are not present in the lanes loaded with GST-CycA alone: what are these? A lane loaded with WRN alone is required.

The Figure 1C has been moved to supplementary materials and now is the Supplementary Figure 1. A short exposure time of the autoradiography is now included in the figure. We now indicate migration of the GST-CyclinA protein with a white box. Indeed, phosphorylation involves the CyclinA protein, as the signal is detected in all the lanes that contain the CDK2/CycA complex. Other phosphorylated proteins that appear in the long exposure image could be degradation products of GST-CyclinA or GST-CWRN.

The result shown in Fig. 1D is not convincing. The claimed decrease in WRN phosphorylation is barely detectable.

We repeated all the IP/WB experiments supporting *in vivo* CDK-dependent phosphorylation of S1133. The new results, provided in Figure 1, Supplementary Figure 2 and 3, were obtained using the pS1133WRN specific custom antibody and are much more convincing in supporting our conclusions.

An immunoprecipitation with non-specific IgG should also be included to show the specificity of the anti-Flag immunoprecipitation.

As a control, we used cells transfected with the empty vector, thus in cells that do not express any Flag-tagged WRN or other Flag-tagged proteins. From this point of view, our control is equivalent to that asked by the reviewer and represent the standard when analyzing Flag-targets transiently-expressed or stably expressed in cells.

The experiment shown in Fig. 1E is unacceptable under the present form. The samples should be load in the same gel to be compared. If the figure comes from the same cropped gel, then the entire gel should be shown. In particular, this panel shows that the CDK1 inhibitor reduces but does not eliminate WRN phosphorylation, suggesting that kinases other than CDK1 may also phosphorylated WRN on S1133 residue.

We now repeated all the IP/WB experiments supporting *in vivo* CDK-dependent phosphorylation of S1133. The new results, provided in Figure 1, Supplementary Figure 2 and 3, were obtained using the pS1133WRN specific custom antibody and much more convincingly support our claim.

2. The inclusion of untreated (-CPT) cells in Fig. 2B and 3C would strengthen the data. The effect of WRN mutants on the recruitment of RPA (not phosphorylated) at DSBs by immunofluorescence and western blot should be also addressed. This is an important point since RPA is strongly implicated in long end resection in both yeast (Chen et al., Mol Cell 2013) and Xenopus (Tammaro et al., Nucl. Acids Res. 2015), and it is part of the mechanism explored by the authors. These works are not cited in the manuscript.

Images from untreated cells are shown in Supplementary Fig 5A. We now included (Supplementary Figure 5E) the analysis of recruitment of RPA32 performed by IF. The analysis of RPA32 by WB in chromatin was already shown in the previous version of the ms, and can be found now in Supplementary Figure 5D. As already reported, recruitment of RPA32 at ssDNA can be analysed by WB only from chromatin fractions.

We included a statement about the paper of Tammaro et al., which is much more related to our work, in the current version of the ms.

3. Analysis of resection in WS cells with or without WRN inhibitor should be included to show the specificity of the inhibitor. The western blot shown in Fig. 3D is unacceptable under the present form. A whole cell extract with a loading control and WRN blot should be shown to appreciate the efficiency of siRNA EXO1. Then a second panel showing the immunoprecipitation should be included showing the IgGs and WRN levels.

We now included quantification of ssDNA in WS cells in the presence or not of the WRN helicase inhibitor. We provided a new WB from an anti-EXO1 IP in which is also shown the IgG control. Images from low and high exposure times are now included to unambiguously show that the EXO1 RNAi was effective. Unfortunately, the level of EXO1 in the cells is so low that it is impossible to detect the protein by WB from WCEs in a reliable manner. For this reason, all papers showing EXO1 RNAi data, checked actual down-regulation by Q-PCR or by IP/WB. Thus, we cannot provide WBs on WCE as requested by the reviewer.

4. The length of the DNA fibers analyzed in all the experimental conditions must be shown in Figure 4.

We did not understand the reviewer's comment. A graph with the length of IdU tract in the untreated cells was already shown. Does the reviewer refer to the length of the tract in the CPT-treated cells? If so, the aim of the CPT experiment was to evaluate fork restart, and thus only the number of stalled vs. restarted forks has been recorded. We will be happy to provide the reviewer with a graph showing the length of the IdU tracts in the restarting forks after CPT treatment, if needed, but even in its absence the conclusion that loss of WRN-S1133 phosphorylation reduces the number of restarting forks remains valid.

5. A blot with WRN should be shown in Fig. 5B.

The requested WB has been included in the figure.

6. In Fig. 6C, the claim that Rad51 accumulates onto chromatin upon CPT treatment is unconvincing. First, cytoplasmic and chromatin samples obtained during the fractionation procedure showing an enrichment of Rad51 onto chromatin and of that of chromatin-bound proteins (histones or the Origin Recognition Complex, ORC) should be shown in the same gel. Then, the claimed enrichment of Rad51 must be shown using low exposures of the western blot and quantified against at least two different loading controls.

We now included analysis of the amount of RAD51 in the WCEs to rule out that differences of the protein accumulation in the chromatin fraction could derive from differences in the total level of the protein.

We also repeated WB using ORC2 to normalize against two loading controls, as requested. (See Supplementary Fig. 14)

Fig. 6D, the sentence « in wild-type cells, the level of DSBs induced by CPT dropped of more than 50% after recovery » does not sustain the observed data shown in Fig. 6D, where a level lower than 50% is observed. However this appear to be the case in the phosphomimetic mutant. Also, this latter does not seem to stimulate significantly repair after recovery when compared to cells complemented with wild-type WRN. This must be corrected and the interpretation of these data must be revised.

We amended the sentence and make the point clearer than it could be in the original version of the ms. For what concerns repair stimulation in the WRN-S1133D-expressing cells, our claim was correct: in wild-type cells, DSBs decrease of about the 30% at 4h, while they decrease of more than 50% in the S1133D mutant. Differences are lost at 19h of recovery. We now better explained our result.

7. Materials and Methods. The reference of the antibodies used in the manuscript must be indicated. The experimental procedure to produce recombinant WRN and CDK2/Cyc-A complex is not described.

We included a table with all the antibodies used, dilutions and vendor, in supplementary material. The procedure used to purify GST-CWRN has been included in the supplementary materials and methods. CDK2/CycA is a commercial reagent (see supplementary materials and methods).

G. Previous work is correctly cited except for the following two papers that have not been cited showing an implication of RPA with WRN both in yeast and Xenopus respectively: Chen et al., Mol Cell 2013, Tamaro et al., Nucl. Acids Res. 2015, .

As anticipated in one of our previous answers, we included citation to the work of Tamaro et al., which contains information obtained in a model that more closely fits with the focus of our work.

H. The paper is clearly and lucidly written, making the wright point on the main issue of the work. The writing can be improved in several places by correcting several spelling mistakes upper versus lower case for gene names, using appropriate expressions for describing experimental procedures and data, and explaining the experimental set up. The way the paper is written is too specialized (see some suggestions below). I suggest the paper to be proofread by an english native speaker.

We carefully proofread the ms and had it checked by one of the coauthors, who is an English native speaker.

The authors should explain that WS cell line is mutated in WRN, which is fundamental to interpret the results.

We included this information in materials and methods as well as at the beginning of the relevant paragraph in the results section.

Page 5, line 9, the sentence "...treated cultures with nothing.." » should be changed to « untreated cultures » or to a similar sentence.

Page 7, fourth line of the second paragraph, « after 30 minutes of CPT, consistently... » change to « after 30 minutes of CPT treatment, consistent... ». Last paragraph, first line, change « the WRN helicase activity supports the DNA2

exonuclease... » to « WRN helicase stimulates DNA2 exonuclease activity ... ». Line 6 change « consistently ... » to « consistent... ».

The title and the first line of the last paragraph of page 8 are unclear.

As exemplified by the reviewer, errors in the structure of the sentences have been now corrected.

Page 9, first line, the significance of the experimental set up must be clearly described (why replication forks are pulse-abeled with IdU/CdU ?). Last line, please explain what PLA is. The same must be also done for the last paragraph of page 10. Please explain the meaning of detection of 53BP1 foci.

We included the rational of the experiment in the relevant section of the results. Also, we included a reference for the PLA technique. For what concerns detection of the 53BP1 foci, this experiment is no more included in the revised version of the ms, as suggested by reviewer #2.

Page 11, explain the rational of the COMET assay.

The rational of the comet assay was already in the original version of the ms. We tried to make it clear in the revised text.

Discussion, "change unphosphorylable" to "unphosphorylatable".

We noticed that both words are used and can be found in published papers.

The effect of WRN S1133A mutant on the recruitment of BLM suggested in the discussion can be shown experimentally.

This is an interesting point. We checked BLM foci in WS, WRN-WT and WRN-S1133A cells. Actually, WS cells show more BLM-foci-positive nuclei than WT or WRN-S1133A cells. We included the results of the experiment in the revised version of the ms as Supplementary Figure 10.

Supplementary material for reviewer #1 and #3

Inhibition of the proteasome by MG132 does not stabilize WRN protein. WS cells complemented with the wild-type, unphosphorylatable or phosphomimetic WRN form have been cultured in the presence or not of 25 μ M MG132 for 4h, which corresponds to the longer treatment time at which ssDNA was analysed in the study. Thereafter, a WCE was prepared and proteins analysed by SDS-PAGE and WB using anti-WRN and anti-tubulin antibodies.

REVIEWERS' COMMENTS:

Reviewer #1 (Remarks to the Author):

The authors have significantly improved the manuscript, in terms of quality of data and presentation. They provided many new, well-done experiments collectively requested by the previous reviewers. They now convincingly demonstrate that Cdk phosphorylation of WRN on S1113 is important for end resection and repair pathway choice at replication coupled DSBs. They prepared phosphopeptide specific antibodies that convincingly demonstrate phosphorylation in vivo. They purified non-phosphorylatable S1113 mutant proteins, revealing that catalytic activities were intact, increasing confidence that effects of the mutations are due to lack of phosphorylation rather than a change in conformation or stability. They showed that the mutant proteins were stable. They improved controls suggesting that phosphorylation is important for interaction with Mre11. Finally, they improved the data suggesting that NHEJ is increased in S1113A and, by demonstrating increased SCE, that hyper-recombination is occurring. The text was significantly clarified by re-writing.

The results are significant because they reveal for the first time that WRN is an additional regulatory target of Cdk in resection and that inhibition of phosphorylation leads to significant changes in pathway choice at replication induced DSBs. Therefore, the manuscript is now suitable for publication.

Reviewer #2 (Remarks to the Author):

The authors have satisfied my previous concerns and have strengthened the study with new data and experiments. The antibody against WRN phosphorylated at residue S1133 is a valuable new resource. It would be good to demonstrate that this antibody does not stain extracts from Werner syndrome cells. This would further ensure specificity.

Point-by-point response to reviewers

Reviewer #2

We thank the reviewer for the appreciation of our work and to have contributed to improving it with the insightful suggestions. We now included in Supplementary Figure 3C, an IP/WB experiment from WS cells and WS cells complemented with the Flag-tagged WRN wild-type to further confirm specificity of the anti-pS1133WRN antibody.